# A PRISMA systematic review of adolescent gender dysphoria literature: 2) mental health

Lucy Thompson [1,2,3]*, Darko Sarovic [1], Philip Wilson [3], Angela Sämfjord [4], Christopher Gillberg [1,2]

**1** Gillberg Neuropsychiatry Centre (GNC), University of Gothenburg, Göteborg, Sweden, **2** Institute of Health and Wellbeing, University of Glasgow, Glasgow, United Kingdom, **3** Institute of Applied Health Science, University of Aberdeen, Centre for Health Science, Inverness, United Kingdom, **4** The Child and Adolescent Psychiatric Clinic, The Queen Silvia Children's Hospital, Gothenburg, Sweden

* Lucy.Thompson@gnc.gu.se

**Data Availability Statement:** All data are provided within the article.

**Funding:** The author(s) received no specific funding for this work.

## Abstract

It is unclear whether the literature on adolescent gender dysphoria (GD) provides sufficient evidence to inform clinical decision making adequately. In the second of a series of three papers, we sought to review published evidence systematically regarding the extent and nature of mental health problems recorded in adolescents presenting for clinical intervention for GD. Having searched PROSPERO and the Cochrane library for existing systematic reviews (and finding none), we searched Ovid Medline 1946 –October week 4 2020, Embase 1947–present (updated daily), CINAHL 1983–2020, and PsycInfo 1914–2020. The final search was carried out on the 2nd November 2020 using a core strategy including search terms for 'adolescence' and 'gender dysphoria' which was adapted according to the structure of each database. Papers were excluded if they did not clearly report on clinically-likely gender dysphoria, if they were focused on adult populations, if they did not include original data (epidemiological, clinical, or survey) on adolescents (aged at least 12 and under 18 years), or if they were not peer-reviewed journal publications. From 6202 potentially relevant articles (post deduplication), 32 papers from 11 countries representing between 3000 and 4000 participants were included in our final sample. Most studies were observational cohort studies, usually using retrospective record review (21). A few compared cohorts to normative or population datasets; most (27) were published in the past 5 years. There was significant overlap of study samples (accounted for in our quantitative synthesis). All papers were rated by two reviewers using the Crowe Critical Appraisal Tool v1·4 (CCAT). The CCAT quality ratings ranged from 45% to 96%, with a mean of 81%. More than a third of the included studies emerged from two treatment centres: there was considerable sample overlap and it is unclear how representative these are of the adolescent GD community more broadly. Adolescents presenting for GD intervention experience a high rate of mental health problems, but study findings were diverse. Researchers and clinicians need to work together to improve the quality of assessment and research, not least in making studies more inclusive and ensuring long-term follow-up regardless of treatment uptake. Whole population studies using administrative datasets reporting on GD / gender non-conformity may be necessary, along with inter-disciplinary research evaluating the lived experience of adolescents with GD.

**Competing interests:** The authors have declared that no competing interests exist.

# Introduction

This is the second of three papers examining the literature on adolescent Gender Dysphoria (GD) (see Thompson et al [1] for paper 1, paper 3 in preparation). Some sections of the introductory and methodological text, and reference to methodological limitations, are necessarily repeated across all three papers. The definitions and terminology used in paper 1 were also used in the present paper [1].

Gender Dysphoria (GD) is a categorical diagnosis in the Fifth Edition of the Diagnostic and Statistical Manual of Mental Disorders (DSM-5) [2]. It is also used as a general descriptive term referring to a person's discontent with assigned gender. In recent years, GD diagnoses have been increasingly made in child and adolescent services [3–5]. There has been a parallel increase in demand for gender transition interventions, particularly among natal females [3–5]. Current clinical guidance for gender transition in adolescence follow the so-called 'Dutch model', where intervention is staged in accordance with a young person's age and stage of pubertal development [6,7]. The age at which a stage of intervention will be deemed appropriate is based partly on how reversible it is. The first stage, puberty suppression (prevention of the development of secondary sexual characteristics), is reversible (although not without risks to health and wellbeing) [8], the second stage, cross-sex hormone treatment, is reversible to some extent (although there is a lack of evidence regarding its longer term impact) [8], and the third stage, surgical intervention, is irreversible. Consideration of a young person's mental health status is an important component of the assessment process. A young person must be deemed suitably competent to make treatment decisions, as well as suitably distressed to warrant intervention in the first place. There is a need, therefore, to have a good understanding of mental health profiles within this population.

There is now a substantial literature showing that adolescents with GD experience poor mental wellbeing in comparison to their peers [9,10], including suicidal thoughts and behaviour [11,12], and some evidence that mental wellbeing improves for those taking part in intervention programs [6,13,14]. There is inherent bias in some of these studies, however, for example several papers rely on survey data with no record of natal sex, others acknowledged lack of socioeconomic representation in samples, and systematic reviews on the subject have not focused on adolescence [15,16]. Therefore, questions remain about the place of GD within the broader context of a young person's mental health. There is evidence that most prepubertal children with GD desist once they reach puberty [17], whereas adults are more likely to persist [18]. There is some indication that adolescents are unlikely to desist, but there is a lack of relevant recent follow-up studies [19,20]. There is an explicit lack of evidence on adolescent-onset GD, so the interplay between GD and other MH factors in this phenomenon is not well understood [21].

Intense international debate regarding a number of issues relating to GD in adolescence is ongoing, especially within Europe and North America where the main research active treatment centres are based [21]. One recent high profile legal case (Bell vs Tavistock [22]) attracted considerable attention from people and organisations with a range of strongly-held views both in favour of and against the ruling, illustrating the acknowledged lack of good quality evidence regarding treatment comorbidities and outcomes to inform service design [23,24]. Services are sometimes left having to make unilateral decisions against national guidance, e.g., the Karolinska University Hospital in Stockholm, Sweden, changing their policy to limit puberty suppression to the context of clinical studies [25].

## Scope of the review

This review is the second in a three-part series addressing the current state of evidence on gender dysphoria experienced in adolescence. Our over-arching aim was to establish what the

literature tells us about gender dysphoria in adolescence. We broke this down into seven specific questions (see below). Paper 1 [1] addressed questions 1-3c (italicised), Paper 2 (the current paper) addresses question 4 (bold text), and Paper 3 will address questions 3d and 5–7 (plain text).

1. *What is the prevalence of GD in adolescence*?

2. *What are the proportions of natal males / females with GD in adolescence (a) and has this changed over time (b)*?

3. *What is the pattern of age at (a) onset (b) referral (c) assessment* (d) treatment?

4. **What is the pattern of mental health problems in this population?**

5. What treatments have been used to address GD in adolescence?

6. What outcomes are associated with treatment/s for GD in adolescence?

7. What are the long-term outcomes for all (treated or otherwise) in this population?

The present paper focuses on question 4. We have addressed questions 1, 2, 3a, 3b, and 3c in our first paper [*ref*], and questions 3d, and 5–7 in a final paper (in preparation). The methodology below includes the searches conducted for the whole review.

We set out to include any paper offering primary data in response to any of these questions.

## Methods

### Protocol and registration

The systematic review protocol was submitted to PROSPERO on the 28th November 2019, and registered on 17 March 2020 (registration number CRD42020162047). An update was uploaded on 2nd February 2021 to include specific detail on age criteria and clinical verification of condition. The review has been prepared according to PRISMA 2020 [26] guidelines (see S1 Checklist).

**Eligibility criteria.** The volume of non-peer-reviewed literature in initial searches proved so great that we took the decision to only include peer-reviewed journal papers featuring original research data. This decision was made subsequent to initial PROSPERO registration, but prior to full text screening. Complete inclusion criteria were:

- Focused on gender dysphoria or transgenderism;

- Includes data on adolescents (aged 12–17 years inclusive);

- Includes original data (not review paper or opinion piece);

- Peer-reviewed publication (not theses or conference proceedings);

- In English language.

### Information sources

We searched PROSPERO and the Cochrane library for existing systematic reviews. We searched Ovid Medline 1946 –October week 4 2020, Embase 1947–present (updated daily), CINAHL 1983–2020, and PsycInfo 1914–2020. After selecting the final sample of articles, the first author used their reference lists as a secondary data source.

## Search

The final search was carried out on 2nd November 2020 using a core strategy which was adapted according to the structure of each database. The core strategy included search terms for 'adolescence' and 'gender dysphoria'. This was kept deliberately broad in order to ensure any studies on the subject could be screened for eligibility. The specific search strategy employed in EMBASE is given below, and represents the format followed with the others. The specific search strategies employed in each database are detailed in Table 1.

EMBASE search

1. Exp adolescence/

2. (adolesc* or teen* or puberty*).tw.

3. 1 or 2

4. exp gender dysphoria/

5. exp transgender/

6. sex reassignment/

7. (gender dysphoria or gender identity or transsex* or trans sex or transgender or trans gender or sex reassignment).tw.

8. 4 or 5 or 6 or 7

9. 3 and 8

**Table 1. Search terms.**

| | EMBASE | Ovid Medline | CINAHL | PsycInfo |
|---|---|---|---|---|
| **Adolescence** | 1. Exp adolescence/<br>2. (adolesc* or teen* or puberty*).tw. | 1. Exp adolescence/<br>2. (adolesc* or teen* or puberty*).tw. | 1. (MH "Adolescence+")<br>2. TI adolesc* OR TI teen* OR TI pubert* OR AB adolesc* OR AB teen* OR AB pubert* | 1. TI adolescence OR AB dolescence<br>2. TI adolesc* OR TI teen* OR TI pubert*<br>3. AB adolesc* OR AB teen* OR AB pubert* |
| **Gender Dysphoria** | 3. exp gender dysphoria/<br>4. exp transgender/<br>5. sex reassignment/<br>6. (gender dysphoria or gender identity or transsex* or trans sex or transgender or trans gender or sex reassignment).tw. | 3. exp gender dysphoria/<br>4. exp transgender/<br>5. Exp Sex Reassignment Procedures/<br>6. (gender dysphoria or gender identity or transsex* or trans sex or transgender or trans gender or sex reassignment).tw. | 3. (MH "Gender Dysphoria")<br>4. (MH "Transgender Persons") OR (MH "Transsexuals")<br>5. (MH "Sex Reassignment Procedures+")<br>6. TI gender dysphoria OR AB gender dysphoria OR TI gender identity disorder OR AB gender identity disorder OR TI transsex* OR AB transsex* OR TI trans sex* OR AB trans sex* OR TI transgender OR AB transgender OR TI trans gender OR AB trans gender<br>7. TI sex reassignment OR AB sex reassignment OR TI gender reassignment OR AB gender reassignment | 4. DE "Gender Dysphoria" OR DE "Gender Nonconforming" OR DE "Gender Reassignment" OR DE "Gender Identity" OR DE "Transsexualism" OR DE "Transgender"<br>5. TI gender dysphoria OR TI gender identity disorder OR TI transsex* OR TI trans sex* OR TI transgender OR TI trans gender OR TI sex reassignment OR TI gender reassignment<br>6. AB gender dysphoria OR AB gender identity disorder OR AB transsex* OR AB trans sex* OR AB transgender OR AB trans gender OR AB sex reassignment OR AB gender reassignment |
| **Combination terms** | 7. 1 OR 2<br>8. 3 OR 4 OR 5 OR 6<br>9. 7 AND 8 | 7. OR 2<br>8. 3 OR 4 OR 5 OR 6<br>9. 7 AND 8 | 8. 1 OR 2<br>9. 3 OR 4 OR 5 OR 6 OR 7<br>10. 8 AND 9 | 7. 1 OR 2 OR 3<br>8. 4 OR 5 OR 6<br>9. 7 AND 8 |

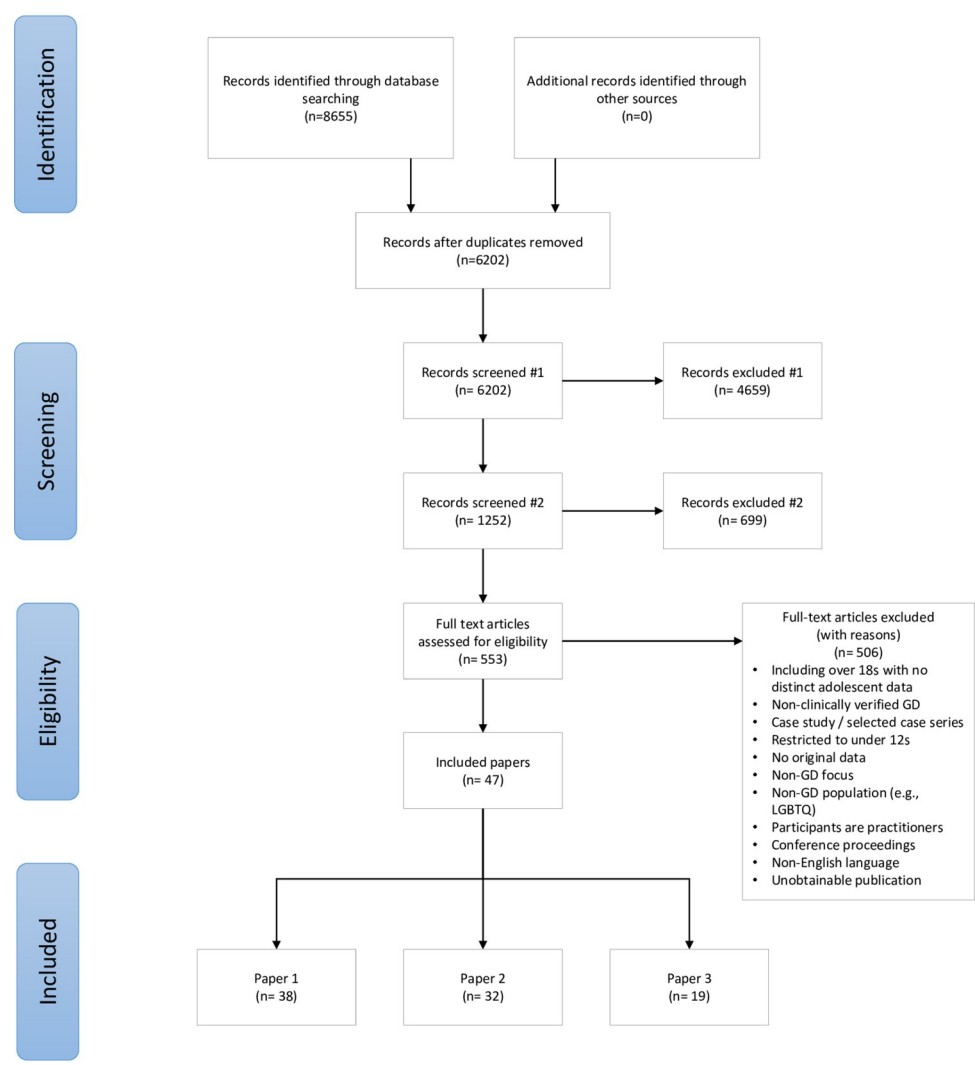

**Fig 1. PRISMA diagram.**

## Study selection

The study selection process is illustrated in Fig 1. We used Endnote v. X7.8 to manage all references, and followed the de-duplication and management strategies set out in Bramer et al (2016) [27] and Peters (2017) [28] respectively.

In the first stage of screening, papers were excluded based on their title or abstract if they did not clearly report on gender dysphoria or transgenderism and if they were focused on adult populations. In the second stage of screening, papers were excluded on the basis of title and abstract if they did not include original data (epidemiological, clinical, or survey) on adolescents (aged at least 12 and under 18 years). At both stages papers were retained if there was insufficient information to exclude them.

Full-text files were obtained for the remaining records.

Papers were rejected at this stage if they:

- Contained no original data (including literature and clinical reviews, journalistic / editorial pieces, letters and commentaries);

- Included only case studies or selected case series;

- Pertained to conditions other than GD (e.g., Disorders of sexual development or HIV);

- Did not include clinically-identified GD (e.g., survey where participants self-identify, with no clinical contact);

- Pertained to populations other than those with GD (e.g., LGBTQ more broadly);

- Pertained to populations including or restricted to those aged 18 years or older. This included papers where adolescents and adults were included in the same sample, but adolescents were not separately reported (in many cases age range was not reported and so a 'balance of probabilities' assessment had to be made based on the reported mean age);

- Pertained to populations restricted to those aged under 12 years of age. This included papers where adolescents and children were included in the same sample, but the majority of participants were clearly under 12 (based on mean or median age);

- Where participants were practitioners, not patients;

- Referred only to conference proceedings;

- Were written in a non-European language (e.g., Turkish);

- Could not be obtained (including due to being published in non-English language journals, or in theses).

Following initial full text screening, all remaining papers were assessed by a second reviewer to reduce the risk of inclusion bias. Where reviewers reached a different conclusion, discussion took place to reach consensus. If agreement could not be reached, a third reviewer was consulted, and discussion used to reach consensus amongst all three reviewers.

Data extracted from eligible papers were tabulated and used in the qualitative synthesis. Given the limited number of specialist treatment centres globally, we assessed how many of the included papers featured the same or overlapping samples.

Papers included in the sub-sample for the present analysis contained some indication of mental health (MH) status at assessment / pre-intervention / baseline for adolescents experiencing (clinically likely) GD. Although some papers reported on bullying or school victimisation, we have omitted these findings as, whilst they may represent a risk of MH problems, they are not symptoms in and of themselves.

### Quality assessment

All papers were rated by two reviewers using the Crowe Critical Appraisal Tool v1·4 (CCAT [29]). CCAT is suitable for a range of methodological approaches, assessing papers in terms of eight categories: Preliminaries (overall clarity and quality); Introduction; Design; Sampling; Data collection; Ethical matters; Results; Discussion. Each category is rated out of 5 and all eight categories summed to give a total out of 40 (converted to a percentage). In the present review, each paper was then assigned to one of five categories, based on the average rating of the reviewers, where a rating of 0–20% was coded 1 (poorest quality), and 81–100% coded 5 (highest quality). Inter-rater reliability was shown to be very good ($k$ = 0·92, SE = 0·05).

### Data collection process

Data were extracted from the papers using the CCAT form (https://conchra.com.au/wp-content/uploads/2015/12/CCAT-form-v1.4.pdf) by two reviewers per paper and compiled by

the first author (LT). Once compiled, instances of overlap between papers (i.e., if the same sample was described in two papers) were identified and tabulated, and the final sample for each question defined.

## Results

### Number of studies included, retained and excluded

The PRISMA diagram in Fig 1 provides details of the screening and exclusion process. The searches returned 8655 results, reduced to 6202 following de-duplication. Titles and abstracts were screened by one reviewer (LT) and 4659 records excluded after initial screening and a further 699 excluded on second stage title / abstract screening. This left 553 eligible for full text screening. An initial screening (LT) of full texts reduced the number of records to 155. Forty-seven papers were included in the final dataset, of which 32 included data for the present paper. Full characteristics of included studies are provided in Table 2.

Most papers (n = 25) had a focused research question pertaining to MH; others set out to measure MH status as part of the assessment process, while others included any available MH data as part of their characterisation of samples. Four papers were interested explicitly in autism symptoms, and one focused on eating disorder symptoms.

Clusters of samples came from the same regions, specifically in the Netherlands (n = 6), the UK (n = 6), and Canada (n = 5). The USA had the highest number of samples (n = 8), with the remainder from the following countries: Belgium (n = 3), Finland (n = 2), Germany (n = 2), Italy (n = 1), Switzerland (n = 1), Australia (n = 1), and Turkey (n = 1) (note two papers together described six samples, hence the total is 36). The Netherlands data all pertained to the same centre and research group. All six of the UK samples came from the same Gender Identity Development Service (GIDS: Tavistock & Portman NHS Trust) in London, three of the Canadian papers came from the same Transgender Youth Clinic in Toronto, three of the US samples, all three Belgian samples, both Finnish samples, and both German samples each came from the same centres. Accordingly, not all 36 samples are necessarily mutually exclusive. Overlapping samples were not always acknowledged, and so where overlap has / may have occurred (based on location, setting, age and date variables) this has been noted and has been taken into account in any analysis. We generally aimed to include the largest sample / sample covering the widest date range, to optimise representativeness, and sought to avoid 'double counting' where samples clearly overlapped. Fig 2 provides a graphical representation of likely overlap between samples. Based on the reported information, in total we estimate between 3000 and 4000 adolescents assessed at specialist centres for GD between 1980 and 2020 were included in the 32 papers.

Most studies were observational cohort studies, usually using retrospective record review (n = 21). A few studies included a comparison group / groups from another GD clinic (n = 2), or from a non-GD population (n = 5). All but one paper was published within the past ten years (2011 or later) and all but five in the past five years (2016 or later). Only two papers explicitly included data from before 2000 (a further six may have included pre-2000 data but did not report dates). All but two papers included both natal male (NM) and natal female (NF) participants (Tack et al (2016) NF only; Tack et al (2017) NM only). All studies reported the proportion of NM and NF participants in their sample (see Table 2 and Fig 2).

The means of assessing mental health status varied across the studies, with self-report (n = 18) and parent-report (n = 17) measures being by far the most common. Participant-reported psychiatric history at clinic intake (n = 7), or history from medical notes (n = 7) were also used, and measures requiring clinician assessment reported in a further seven. In 17/36 of

**Table 2. Study characteristics.**

| ID | Country | Reference | Design | Setting | N | Age (years) | Male natal sex (%) | Date range | GD status | MH status |
|----|---------|-----------|--------|---------|---|-------------|--------------------|------------|-----------|-----------|
| 1 | Australia | Mahfouda, Panos, et al. (2019). Mental Health Correlates of Autism Spectrum Disorder in Gender Diverse Young People: Evidence from a Specialised Child and Adolescent Gender Clinic in Australia. Journal of Clinical Medicine, 8(10), 20 [30] | obs, retro, x-sect | Gender Diversity Service (GDS), Perth Children's Hospital. GENTLE cohort study | 104 | 14·6±1·7 | 24·0 | Nov 2017-Jun 2019 | 4 | 1 2 3 |
| 2 | Belgium | Tack, et al. (2016). Consecutive lynestrenol and cross-sex hormone treatment in biological female adolescents with gender dysphoria: A retrospective analysis. Hormone Research in Paediatrics, 86 (Supplement 1), 268–269 [31]. | obs, retro, longit, interv | Division of Pediatric Endocrinology, Ghent University* | 45 NF | mean age start of progestins 15.8 | 0 | 2010–2015 | 1 | 3 |
| 3 | Belgium | Tack, L. J. W., Heyse, R., Craen, M., Dhondt, K., Bossche, H. V., Laridaen, J., & Cools, M. (2017). Consecutive Cyproterone Acetate and Estradiol Treatment in Late-Pubertal Transgender Female Adolescents. Journal of Sexual Medicine, 14(5), 747–757 [32] | obs, retro, longit, interv | Division of Pediatric Endocrinology, Ghent University*, Belgium | 27 NM | Mean age at start of CA: 16y 6m, CA +estradiol: 17y 7m | 100 | 2008–2016 | 1 | 3 |
| 4 | Canada | Chiniara, et al. (2018). Characteristics of adolescents referred to a gender clinic: Are youth seen now different from those in initial reports? Hormone Research in Paediatrics, 89(6), 434–441 [33] | obs, retro, x-sect | Transgender Youth Clinic (TYC), The Hospital for Sick Children, Toronto | 203 | 12–18 (mean 16) | 23·2 | 2014–2016 | 1 | 2 3 |
| 5 | Canada | Feder, et al. (2017). Exploring the association between eating disorders and gender dysphoria in youth. Eating Disorders, 25 (4), 310–317 [34] | obs, retro, x-sect | Gender Diversity Clinic at a Canadian tertiary pediatric care hospital in Ottawa, Ontario | 97 | 12–18 | 38·1 | Oct 2007-Jul 2015 | 1 | 3 |
| 6 | Canada | Heard, et al. (2018). Gender dysphoria assessment and action for youth: Review of health care services and experiences of trans youth in Manitoba. Paediatrics & Child Health (1205–7088), 23(3), 179–184 [35] | obs, retro, x-sect | Manitoba Gender Dysphoria Assessment and Action for Youth (GDAAY) program | 174 | 4·7–17·8 | 29·9 | None given | 2 | 4 |
| 7 | Canada | Sorbara, et al. (2020). Mental Health and Timing of Gender-Affirming Care. Pediatrics, 146 (4), 10 [36] | obs & comp, retro, x-sect | Transgender Youth Clinic (TYC), The Hospital for Sick Children, Toronto | 300 | 10·5–17·9 | 23·7 | initial visit Oct 2013—Jun 2016 (cohort 1) or Aug 2017—Jun 2018 (cohort 2) | 1 | 3 |

(*Continued*)

**Table 2.** (Continued)

| ID | Country | Reference | Design | Setting | N | Age (years) | Male natal sex (%) | Date range | GD status | MH status |
|----|---------|-----------|--------|---------|---|-------------|--------------------|-----------|-----------|-----------|
| 8 | Finland | Kaltiala-Heino, et al. (2015). Two years of gender identity service for minors: Overrepresentation of natal girls with severe problems in adolescent development. Child and Adolescent Psychiatry and Mental Health, 9 (1) [37] | obs, retro, x-sect | Tampere University Hospital, Department of Adolescent Psychiatry | 47 | mean NM 16·04±0·57, NF 16·66 ±1·07 | 12·8 | 2011–2013 | 2 | 4 |
| 9 | Finland | Kaltiala-Heino, et al. (2019). Sexual experiences of clinically referred adolescents with features of gender dysphoria. Clinical Child Psychology and Psychiatry, 24(2), 365–378 [38] | obs, retro, x-sect | Tampere University Hospital, Department of Adolescent Psychiatry | 99 | 14–18 | 15·2 | 2011–2015 | 2 | 4 |
| 10 | Germany | Becker-Hebly, et al. (2020). Psychosocial health in adolescents and young adults with gender dysphoria before and after gender-affirming medical interventions: a descriptive study from the Hamburg Gender Identity Service. European Child and Adolescent Psychiatry [39] | obs, prosp, longit, interv | Gender Identity Service, University Medical Center Hamburg-Eppendorf, Germany | 54 | 11.21–17.34 | 14·8 | Sept 2013 –Jun 2017 | 1 | 2 5 |
| 11 | Germany | Levitan, et al. (2019). Risk factors for psychological functioning in German adolescents with gender dysphoria: poor peer relations and general family functioning. European Child and Adolescent Psychiatry [40] | obs, prosp, x-sect | Hamburg Gender Identity Service for Children and Adolescents | 180 | Mean 15·5 ±1·4 | 18·9 | Sept 2013-Jun 2017 | 1 | 2 |
| 12 | Italy | Fisher, et al. (2017). Psychological characteristics of Italian gender dysphoric adolescents: a case-control study. Journal of Endocrinological Investigation, 40(9), 953–965 [41] | obs & comp, prosp, x-sect | The Sexual Medicine and Andrology Unit of the University of Florence and the Gender Clinics of Rome, Milan, and Naples University Hospitals | 46 (+ 46 control) | <18 | 43·5 | Sept 2014-Feb 2016 | 1 | 2 |
| 13 | Multi | de Graaf, et al. (2018). Psychological functioning in adolescents referred to specialist gender identity clinics across Europe: A clinical comparison study between four clinics. European Child & Adolescent Psychiatry, 27(7), 909–919 [42] | obs & comp, retro, x-sect | (a) Center of Expertise on Gender Dysphoria (CEGD), Amsterdam, Netherlands; | 252 | 12–18 | 46·0 | 2009–2013 | 4 | 1 2 |
| | | | | (b) Pediatric Gender Clinic, Ghent University Hospital, Belgium; | 71 | | 33·8 | | 4 | 1 2 |
| | | | | (c) Department of Child and Adolescent Psychiatry, University Hospital of Psychiatry Zurich, Switzerland; | 26 | | 35·7 | | 4 | 1 2 |

(*Continued*)

**Table 2.** (Continued)

| ID | Country | Reference | Design | Setting | N | Age (years) | Male natal sex (%) | Date range | GD status | MH status |
|---|---|---|---|---|---|---|---|---|---|---|
| | | | | (d) Gender Identity Development Service, Tavistock and Portman NHS Foundation Trust, London, UK | 610 | | 42·3 | | 4 | 1 2 |
| 14 | Multi | de Vries, et al. (2016). Poor peer relations predict parent- and self-reported behavioural and emotional problems of adolescents with gender dysphoria: a cross-national, cross-clinic comparative analysis. European Child and Adolescent Psychiatry, 25(6), 579–588 [43] | obs, comp, x-sect | 1. Center of Expertise on Gender Dysphoria (CEGD), Amsterdam | 139 | 13–18 | 56·8 | 1996–2008 | 1 | 1 2 |
| | | | | 2. Transgender Youth Clinic (TYC), The Hospital for Sick Children, Toronto, Canada | 177 | 13–18 | 53·1 | 1980–2010 | 4 | 1 2 |
| 15 | Netherlands | Cohen-Kettenis & Van Goozen (2002). Adolescents who are eligible for sex reassignment surgery: Parental reports of emotional and behavioural problems. Clinical Child Psychology and Psychiatry, 7 (3), 412–422 [44] | obs, retro, x-sect | University Medical Centre, Utrecht (moved to VUmc / CEGD in 2002) | CBCL: 29 DISC: 21 | 11–18 | 37·9 52.4 | None given | 1 | 1 5 |
| 16 | Netherlands | de Vries, et al. (2011a). Psychiatric comorbidity in gender dysphoric adolescents. Journal of child psychology and psychiatry, and allied disciplines, 52(11), 1195–1202 [45] | obs, prosp, x-sect | VU University Medical Center, Amsterdam (forerunner to CEGD) | 105 | 10·5–18·0 | 50·5 | Apr 2002-Dec 2009 | 1 | 1 5 |
| 17 | Netherlands | de Vries, et al. (2011b). Puberty suppression in adolescents with gender identity disorder: A prospective follow-up study. Journal of Sexual Medicine, 8 (8), 2276–2283 [46] | obs & comp, prosp, longit | VU University Medical Center, Amsterdam (forerunner to CEGD) | 70 | 11·1–17·0 | 47·1 | 2000–2008 | 1 | 1 2 5 |
| 18 | Netherlands | van der Miesen, et al. (2018). Autistic Symptoms in Children and Adolescents with Gender Dysphoria. Journal of Autism and Developmental Disorders, 48(5), 1537–1548 [47] | obs & comp, prosp, x-sect | Center of Expertise on Gender Dysphoria (CEGD), Amsterdam | 490 | Sample mean: 11·1 | 50·6 | Sample: 2005–2012 | 1 | 1 |
| | | | | Comp grp 1: Dutch schools | 2507 | Comp grp 1 mean: 10·1 | | Comp grp 1: 1996–2000 | | 1 |
| | | | | Comp grp 2: ASD dx Dutch psychiatry clinic. | 196 | Comp grp 2 mean: 10·8 | | Comp grp 2: 1996–2000 | | 1 |
| 19 | Turkey | Akgul, et al. (2018). Autistic Traits and Executive Functions in Children and Adolescents With Gender Dysphoria. Journal of Sex & Marital Therapy, 44(7), 619–626 [48] | obs & comp, prosp, x-sect | Marmara University Pendik Education and Training Hospital's Child and Adolescent Psychiatry Clinic | GD: 25; control: 50 (age- & sex-matched, no psychiatric diagnosis) | 5–17 | 52 | Not given | 1 | 1 |

(*Continued*)

**Table 2.** (Continued)

| ID | Country | Reference | Design | Setting | N | Age (years) | Male natal sex (%) | Date range | GD status | MH status |
|----|---------|-----------|--------|---------|---|-------------|-------------------|------------|-----------|-----------|
| 20 | UK | Costa, et al. (2015). Psychological Support, Puberty Suppression, and Psychosocial Functioning in Adolescents with Gender Dysphoria. Journal of Sexual Medicine, 12(11), 2206–2214 [49] | obs & comp, retro, longit, interv | Gender Identity Development Service, Tavistock & Portman, London | 201 (Control: 169 CAMHS Stockholm cases) | 12–17 | 37·8 | 2010–2014 | 1 | 5 |
| 21 | UK | Holt, V., et al. (2016). Young people with features of gender dysphoria: Demographics and associated difficulties. Clinical Child Psychology and Psychiatry, 21(1), 108–118 [50] | obs, retro, x-sect | Gender Identity Development Service, Tavistock & Portman, London | 218 | 5–17 (separate data on adols) | 37·2 | Jan-Dec 2012 | 2 | 4 |
| 22 | UK | Matthews, et al. (2019). Gender Dysphoria in looked-after and adopted young people in a gender identity development service. Clinical Child Psychology & Psychiatry, 24(1), 112–128 [51] | obs, retro, x-sect | Gender Identity Development Service, Tavistock & Portman, London | 185 | 3–17 | 52·4 | 2009–2011 | 1 | 5 |
| 23 | UK | Russell, et al. (2020). A Longitudinal Study of Features Associated with Autism Spectrum in Clinic Referred, Gender Diverse Adolescents Accessing Puberty Suppression Treatment. Journal of Autism and Developmental Disorders [52] | obs, retro, x-sect | Gender Identity Service, Tavistock & Portman, London | 95 | 9·9–15·9 (mean 13·6 ±0·11) | 40 | Not given | 2 | 1 |
| 24 | UK | Skagerberg, et al. (2013). Internalizing and externalizing behaviours in a group of young people with gender dysphoria. International Journal of Transgenderism, 14(3), 105–112 [53] | obs, x-sect | Gender Identity Development Service (GIDS), Tavistock & Portman, London | 141 | 12–18 | 40 | None given | 1 | 2 |
| 25 | USA | Becerra-Culqui, et al. (2018). Mental health of transgender and gender nonconforming youth compared with their peers. Pediatrics, 141 (5) (e20173845) [54] | obs & comp, retro, x-sect | California and Georgia, US, Kaiser-Permanente records | 1082 (adol sample) (control: 21,317 cis gender matched (≥7 per index case)) | 10–17 | 39·5 | 1/1/06–31/12/14 | 3 | 4 |
| 26 | USA | Chen, et al. (2016). Characteristics of Referrals for Gender Dysphoria over a 13-Year Period. Journal of Adolescent Health, 58(3), 369–371 [55] | obs, retro, x-sect | paediatric endocrinology clinic, Indiana | 38 | Mean age 14·4±3·2 | 42·1 | 1/1/02–1/4/15 | 3 | 4 |
| 27 | USA | Edwards-Leeper, et al. (2017). Psychological profile of the first sample of transgender youth presenting for medical intervention in a US pediatric gender center. Psychology of Sexual Orientation and Gender Diversity, 4(3), 374–382 [56] | obs, retro, x-sect | The Gender Management Service (GeMS) program at Boston Children's Hospital | 56 | 8·9–17·9 | 53·6 | 2007–2011 | 4 | 1 2 |

(*Continued*)

**Table 2.** (Continued)

| ID | Country | Reference | Design | Setting | N | Age (years) | Male natal sex (%) | Date range | GD status | MH status |
|---|---|---|---|---|---|---|---|---|---|---|
| 28 | USA | Kuper, et al. (2019a). Exploring the gender development histories of children and adolescents presenting for gender affirming medical care. Clinical Practice in Pediatric Psychology, 7(3), 217–228 [57] | obs, retro, x-sect | 'gender affirming service', University of Texas Southwestern Medical Center* | 224 | 6–17 (6.2% under 10) | 39·4 | 2014–2018 | 1 | 2 |
| 29 | USA | Kuper, et al. (2019b). Baseline Mental Health and Psychosocial Functioning of Transgender Adolescents Seeking Gender-Affirming Hormone Therapy. Journal of developmental and behavioral pediatrics: JDBP., 03 [58] | obs, prosp, x-sect | Gender Education and Care Interdisciplinary Support program, Texas | 149 | 12–18 | 35·6 | 2014–2017 | 1 | 1 2 |
| 30 | USA | Kuper, et al. (2020). Body Dissatisfaction and Mental Health Outcomes of Youth on Gender-Affirming Hormone Therapy. Pediatrics, 145(4), 04 [59] | obs, prosp, longit, interv | 'a multidisciplinary program in Dallas, Texas' | 148 | 9–18 | 37 | Initially assessed 2014–2018 | 1 | 2 5 |
| 31 | USA | Moyer, et al. (2019). Using the PHQ-9 and GAD-7 to screen for acute distress in transgender youth: findings from a pediatric endocrinology clinic. Journal of Pediatric Endocrinology & Metabolism, 32(1), 71–74 [60] | obs, retro, x-sect | 'a pediatric endocrinology clinic', Portland, Oregon | 194 | 11–18 | 24·2 | None given | 2 | 2 |
| 32 | USA | Nahata, et al. (2017). Mental Health Concerns and Insurance Denials Among Transgender Adolescents. LGBT Health, 4 (3), 188–193 [61] | obs, retro, x-sect | 'gender program' at an 'urban, Mid-western, pediatric academic center', Nationwide Children's Hospital, Columbus, Ohio* | 79 | 9–18 | 35·4 | 2014–2016 | 3 | 3 4 |

Key:

* = author's affiliation. No specific setting given obs = observational comp = comparative.

$ = authors report sample overlap prosp = prospective retro = retrospective.

* = inferred—not overtly stated longit = longitudinal x-sect = cross-sectional.

€ = age of first evidence of transgender status in medical records interv = intervention study

a = Max age of whole sample 18.8. Only data on those under 18 years used in this review.

b = 2016 data not used in this review as included participants over 18 years.

GD status codes: 1) clinical diagnosis using DSM-III / IV / IV-TR / 5; 2) active patients within clinic; 3) data mined using ICD 9 / 10 codes and/or relevant keywords; 4) Under assessment at clinic–beyond referral stage.

MH status codes: 1) parent-report measure; 2) self-report measure; 3) psychiatric / MH history reported at intake assessment; 4) psychiatric / MH history from records; 5) clinician assessment / rating.

the samples more than one means of assessing MH status was used, meaning that the remaining 19 studies relied on a single type of measure.

Twenty samples were reported to have met clinical diagnostic criteria for GD / GID, usually using one of the DSM manuals (2/20 did not state which criteria were applied). The remaining 16 samples did not report whether participants met diagnostic criteria, but were included on the basis of being established patients within a specialist treatment centre, either in active

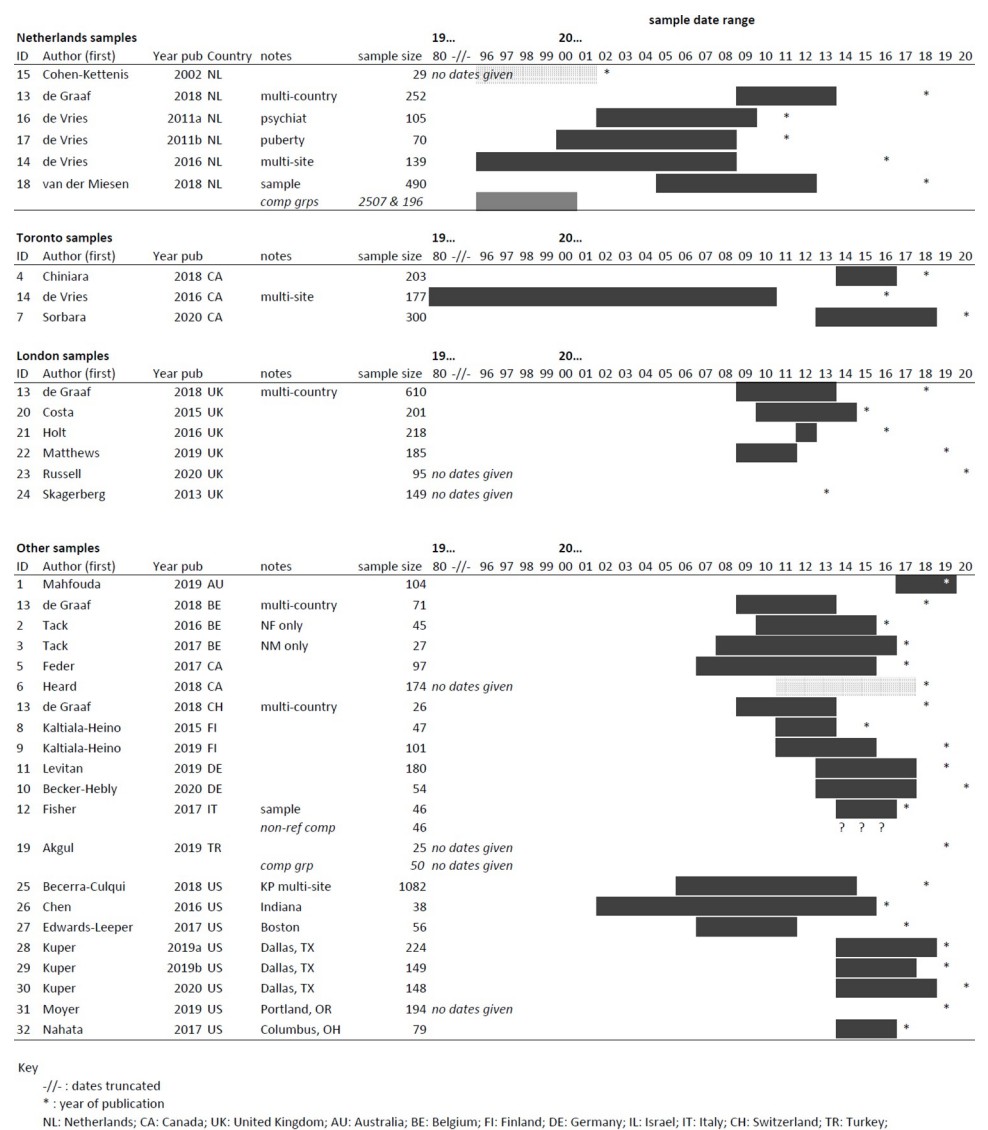

**Fig 2.**

assessment or treatment (n = 13) or were the result of secondary data mining where authors used ICD 9/10 codes and appropriate keywords to establish likely GD (n = 3).

A substantial group of papers narrowly missed inclusion criteria, mostly on the age criterion and some on the clinically likely GD criterion, and were not included in the final sample of reviewed papers. We documented characteristics of all studies excluded at the final full text screen in Table 5.

## Overall findings based on included studies

**What is the pattern of mental health problems in this population?.** Here we present a narrative overview of the findings based on the types of measures employed. Full details of data are in Table 3.

*a) Participant-reported or medical record-recorded psychiatric history.* Previous or concurrent MH diagnoses were common among the included papers. In their large-scale

**Table 3. Quality ratings using Crowe Critical Appraisal Tool (CCAT) [29].**

| ID | Country | Reference | Average CCAT rating[c] | | | | | | | | Total | % | level |
|---|---|---|---|---|---|---|---|---|---|---|---|---|---|
| | | | 1 | 2 | 3 | 4 | 5 | 6 | 7 | 8 | | | |
| 1 | Australia | Mahfouda, et al. (2019). | 4 | 4·5 | 3·5 | 4 | 3·5 | 5 | 4 | 4 | 32·5 | 81 | 5 |
| 2 | Belgium | Tack, et al. (2016). | 4 | 4·5 | 3·5 | 4 | 4·5 | 4·5 | 3·5 | 3 | 31·5 | 79 | 4 |
| 3 | Belgium | Tack, et al. (2017) | 4·5 | 4·5 | 4 | 4 | 4·5 | 5 | 4 | 3·5 | 34 | 85 | 5 |
| 4 | Canada | Chiniara, et al. (2018). | 5 | 5 | 4·5 | 4·5 | 4 | 3·5 | 3·5 | 4 | 34 | 85 | 5 |
| 5 | Canada | Feder, et al. (2017). | 4·5 | 5 | 4·5 | 5 | 4 | 2 | 3·5 | 3·5 | 32 | 80 | 4 |
| 6 | Canada | Heard, et al. (2018). | 4·5 | 5 | 4 | 4 | 3·5 | 2·5 | 3·5 | 4 | 31 | 78 | 4 |
| 7 | Canada | Sorbara, et al. (2020). | 5 | 4 | 4 | 4 | 4 | 2 | 4 | 4·5 | 31·5 | 79 | 4 |
| 8 | Finland | Kaltiala-Heino, et al. (2015). | 4·5 | 5 | 5 | 5 | 4·5 | 4·5 | 4·5 | 5 | 38 | 95 | 5 |
| 9 | Finland | Kaltiala-Heino, et al. (2019). | 5 | 5 | 4·5 | 5 | 4·5 | 5 | 5 | 4·5 | 38·5 | 96 | 5 |
| 10 | Germany | Becker-Hebly, et al. (2020) | 4·5 | 4·5 | 4 | 4·5 | 3·5 | 4 | 4 | 4·5 | 33·5 | 84 | 5 |
| 11 | Germany | Levitan, et al. (2019). | 5 | 5 | 5 | 5 | 4·5 | 4·5 | 5 | 4·5 | 38·5 | 96 | 5 |
| 12 | Italy | Fisher, et al. (2017). | 4 | 4·5 | 4 | 3·5 | 4 | 4 | 4 | 4·5 | 32·5 | 81 | 5 |
| 13 | Multi | de Graaf, et al. (2018). | 5 | 4·5 | 4·5 | 4·5 | 4·5 | 4·5 | 4 | 4 | 35·5 | 89 | 5 |
| 14 | Multi | de Vries et al. (2016) | 4·5 | 4·5 | 4 | 3·5 | 3·5 | 2·5 | 4 | 3·5 | 30 | 75 | 4 |
| 15 | Netherlands | Cohen-Kettenis & Van Goozen (2002). | 2·5 | 2·5 | 3 | 2 | 2·5 | 0 | 3 | 2·5 | 18 | 45 | 3 |
| 16 | Netherlands | de Vries, et al. (2011a). | 5 | 5 | 3·5 | 4·5 | 5 | 4·5 | 4·5 | 3·5 | 35·5 | 89 | 5 |
| 17 | Netherlands | de Vries, et al. (2011b). | 5 | 4 | 2·5 | 3·5 | 4·5 | 4 | 4 | 3 | 30·5 | 76 | 4 |
| 18 | Netherlands | van der Miesen, et al. (2018). | 5 | 5 | 4·5 | 4·5 | 4 | 5 | 4·5 | 4 | 36·5 | 91 | 5 |
| 19 | Turkey | Akgul, et al. (2018). | 4 | 4·5 | 4·5 | 4·5 | 4 | 2·5 | 4 | 4 | 32 | 80 | 4 |
| 20 | UK | Costa, et al. (2015). | 4·5 | 4·5 | 4 | 4·5 | 4 | 3·5 | 4 | 4·5 | 33·5 | 84 | 5 |
| 21 | UK | Holt, V., et al. (2016). | 4 | 4 | 3·5 | 5 | 3·5 | 3·5 | 4 | 3·5 | 31 | 78 | 4 |
| 22 | UK | Matthews, et al. (2019). | 4 | 5 | 4 | 4·5 | 4 | 5 | 3·5 | 4 | 34 | 85 | 5 |
| 23 | UK | Russell, et al. (2020). | 5 | 5 | 3·5 | 3·5 | 4 | 2·5 | 3·5 | 3·5 | 30·5 | 76 | 4 |
| 24 | UK | Skagerberg et al. (2013) | 4·5 | 5 | 4 | 3·5 | 4 | 1 | 4 | 4·5 | 30·5 | 76 | 4 |
| 25 | USA | Becerra-Culqui, et al. (2018). | 4·5 | 4·5 | 4·5 | 4·5 | 4·5 | 4·5 | 4·5 | 4 | 35·5 | 89 | 5 |
| 26 | USA | Chen, et al. (2016). | 4 | 3·5 | 4 | 4 | 4 | 2·5 | 3·5 | 3 | 28·5 | 71 | 4 |
| 27 | USA | Edwards-Leeper, et al. (2017). | 4·5 | 5 | 4·5 | 3·5 | 5 | 3 | 4 | 4·5 | 34 | 85 | 5 |
| 28 | USA | Kuper, et al. (2019a). | 3·5 | 4 | 3·5 | 4 | 4 | 3·5 | 4 | 3·5 | 30 | 75 | 4 |
| 29 | USA | Kuper, et al. (2019b). | 4·5 | 4·5 | 4 | 4 | 3·5 | 4 | 4 | 3·5 | 32 | 80 | 4 |
| 30 | USA | Kuper, et al. (2020). | 5 | 5 | 4 | 4·5 | 5 | 3 | 4 | 3·5 | 34 | 85 | 5 |
| 31 | USA | Moyer, et al. (2019). | 4·5 | 4 | 4·5 | 4·5 | 4 | 4 | 4 | 3·5 | 33 | 83 | 5 |
| 32 | USA | Nahata, et al. (2017). | 4 | 4 | 4 | 4 | 4 | 4 | 4 | 3 | 31 | 78 | 4 |

Key

c 1. Preliminaries (overall clarity and quality); 2. Introduction; 3. Design; 4. Sampling; 5. Data collection; 6. Ethical matters; 7. Results; 8. Discussion

interrogation of administrative datasets Becerra-Culqui et al (2018) reported 71% NM and 74% NF had a MH diagnosis 'ever before index date' and 59% NM and 65% NF in the '6 months before index date'. Depressive disorders were most common, followed by anxiety disorders and attention deficit disorders. The authors noted a particularly increased prevalence rate in psychoses for both NM and NF compared to reference females; for autism spectrum disorders (ASDs) in NM and schizophrenia spectrum disorders in NF compared to reference females; and for suicidal ideation and self-harm in both NM and NF compared to reference males.

Among clinic samples, prevalence of MH problems at baseline assessment ranged from 22% (Tack et al, 2017) to 78% (Sorbara et al, 2020). In a similar profile to that reported in the

large population analysis from Becerra-Culqui et al (2018), mood disorders were most common (depression ranging from 30% to 78%; anxiety from 21% to 63%) and neurodevelopmental disorders were also relatively common (attention deficit/hyperactivity disorder (ADHD) from 6% to 16%; ASD from 2% to 26%). Eating disorders / difficulties were noted in three papers (12% to 15%). Suicidal ideation (12% to 74%) and self-harm (21% to 55%) were also reported, as well as psychotic symptoms (12–13%) and bipolar disorder (5%). Post-traumatic stress disorder (PTSD) was recorded in one study (23%). Between 31% and 37% had been previously or were currently being prescribed psychoactive medication.

*b) Clinician-rated assessment.* Four included studies utilised the Children's Global Assessment Scale (CGAS [62]): a clinician-rated assessment where a single rating (0–100) is given based on a range of information gathered on the young person, including assessments and interviews with parents, children, and school staff; higher scores indicate better global functioning. GD adolescents did not show severe impairment on this scale, scoring either in the 'minor impairment' (71–80), 'some problems' (61–70) (Becker-Hebly et al, 2020; de Vries et al 2011b, Matthews et al, 2019), or 'some noticeable problems' (51–60) (Costa et al, 2015) ranges. Matthews et al (2019) reported that looked after and adopted subgroups scored in a more impaired range than young people living with their birth families, but the samples were small (n = 5 each) in the two former groups, and the differences were not statistically significant. Costa et al (2015) reported GD adolescents had significantly poorer scores than the (no psychiatric symptoms) comparison group, and that NM had more negative mean ratings than NF (although means were both in the same clinical category of 'some noticeable problems'). NM seemed to score more poorly than NF in de Vries et al (2011b) but there was no statistical comparison reported.

Kuper et al (2020) found clinician-reported depressive symptoms (QIDS [63]) were in the 'mild' range, congruent with the self-report version (see below). Scores were significantly higher in NF than NM participants, but authors noted the small effect size (Cohen's $f$ = 0.7) for including gender in the regression model.

*c) Self-report measures.* Most studies (n = 18) used some form of self-report measure. The Youth Self Report (YSR [64]) from the Achenbach System of Empirically Based Assessment (ASEBA [65]) was commonly used as a means of assessing self-reported MH symptoms. Scores on the Total problem scale ranged from 45.7 to 67.1 across included samples, and between 15% and 55% scored 'within the clinical range'. The clinical range was defined in some papers (e.g., T score >63 in Fisher et al, 2017) but not all. Some papers compared their sample to population norms or non-GD samples. In their 2016 paper, de Vries et al reported 41% of participants scoring within the clinical range for Total problems: much higher than in non-referred boys (9%) and girls (8%); Becker-Hebly et al (2020) found YSR scores were significantly worse than published German population norms in adolescents at baseline assessment; and Fisher et al (2017) found their Italian sample of 46 GD adolescents scored higher than non-referred adolescents on Total and Internalising problems (adjusted for age and BMI). Authors observed Internalising problems as being a greater issue than Externalising problems, with some exceptions by natal sex: de Vries et al (2016), de Graaf et al (2018), and Skagerberg et al (2013) each found NM scored significantly higher than NF on Internalising problems, whereas Kuper et al (2019b) found the reverse. Levitan et al (2019) also found female natal sex significantly predicted higher YSR Total problems scores, with de Vries et al (2016) finding NM were more likely to score in the clinical range. Edwards-Leeper et al (2017) and Fisher et al (2017) found no differences by natal sex, and de Vries et al (2011b) did find some differences, but did not present statistical comparisons for all measures at baseline. Some papers were able to compare across regions: in their 2016 paper, de Vries et al compared a Dutch sample with a Canadian sample and found similar percentages of participants scoring within the clinical range. By

contrast, de Graaf et al (2018) found the proportion of their samples falling into the clinical range on the YSR varied significantly between geographical location. Between 15% (Switzerland) and 46% (UK) scored in the clinical range on the YSR (see Table 3). The UK scored most poorly and had the most prevalent emotional and behavioural problems, followed by Belgium and Switzerland.

Moyer et al (2019) reported on simple MH screening tools (PHQ-9 [66]; GAD-7 [67]) used with a sample of adolescents assessed at a paediatric endocrinology unit in the US. At initial consultation, almost half of participants had clinically significant depression symptoms, and over a third had thoughts of death or self-harm (which was significantly higher in non-binary individuals). Almost three quarters of the sample had clinically significant anxiety problems at initial consultation. There were no other differences by gender identity.

In relation to mood disorders, severity of depressive symptoms from self-report measures ranged from average / mild to severe across the samples. Kuper et al (2019a) found about 40% of their sample scored in the moderate to severe range for depression and Chiniara et al (2018) found anxiety levels (as measured by MASC-2 [68]) to be 'significant' in 44% of NF and 30% NM ($p$ = 0.02).

Other concepts measured within our sample include: self-concept (Piers-Harris 2 [69] (Edwards-Leeper, 2017)) which was found to be in the average range; suicide risk (MAST [70] (Fisher, 2017)) which was found to be elevated; body image (BIS [71] (Kuper et al 2019a, 2019b, 2020); BUT [72] (Fisher, 2017)), which was found to be higher than non-referred participants by Fisher et al (2017), but with no comparative data in the Kuper papers; general family functioning (MFAD [73]), where Levitan and colleagues found a third scored above threshold for 'problematic'; and finally quality of life, where scores were worse than population norms on the mental health dimension of the Kidscreen-27 [39,74] and Mahfouda, Panos et al (2019) reported Peds-QL [75] scores to be significantly lower in those GD participants with indicated ASD (compared to the group for whom ASD was not indicated).

*d) Parent-report measures.* Six studies reported on the Child Behavior Checklist (CBCL [76])–the parent-report analogue of the YSR (see above). Reported mean Total problem scores ranged from 44.3 to 64.5 across included samples and between 31% and 55% of samples scored 'within the clinical range', a narrower range than for the YSR. As with the YSR, the clinical range was defined in some papers (e.g., T score >63 in de Vries et al 2011b or score >90th percentile in de Graaf (2018)) but not all. One paper compared their sample to a non-GD sample: de Vries et al (2016) reported 55% of participants scored within the clinical range for Total problems: much higher than in a non-referred sample (9%). As with the YSR, authors observed Internalising problems as being a greater issue than Externalising problems in general, with some exceptions by natal sex: de Graaf et al (2018) found NF scored significantly higher than NM on Externalising problems and there were significantly more NF scoring in the clinical range on Total score; de Vries et al (2011b) also found significantly higher Externalising problem scores in NF compared to NM. Two papers were able to compare across regions: Canadian and UK participants had higher scores and were more likely to have scores in the clinical range than Dutch participants (de Vries et al, 2016; de Graaf et al 2018).

Two studies utilised the DISC-IV [77] to assess psychopathology: Cohen-Kettenis et al (2002) were able to make a diagnosis in 4 of 11 assessed: 1 specific phobia (NF), 1 transient tic disorder (NM), 1 oppositional defiant disorder (NF), 1 overanxious disorder (NM); and de Vries (2011b) reported 32.4% had one or more DSM-IV diagnosis, with anxiety and mood disorders being most prevalent. They also reported that NM were more likely to have two or more diagnoses, mood disorder, or social anxiety disorder than NF (see Table 3).

Akgul et al (2018) reported a GD sample scored higher (more poorly) than a control group on all three domains (metacognitive, behavioural regulation, global executive composite) of the Behavior Rating Inventory of Executive Function (BRIEF [78]).

Three papers used the Social Responsiveness Scale (SRS-2 [79]) to indicate possible ASD, but reported the outcomes in disparate ways: Akgul et al (2018) reported their GD sample scored more highly than controls, and a significantly greater percentage of the GD sample were 'in the clinical range' (but the threshold being used is not clear); Mahfouda, Panos, et al (2019) found 22.1% of their sample to be in the 'severe' clinical range (indicating likely ASD); and Russell et al (2020) report scores in their London sample to be within the mild (T = 60–65) and normal (T≤59) range for NM and NF respectively. Mahfouda, Panos et al (2019) found no differences in SRS-2 score by natal sex, and Akgul et al (2018) reported NF scored significantly higher than NM on SRS social and SRS ADHD-like subscales within the GD group. The Children's Social Behavior Questionnaire (CBSQ [80]) is similar to the SRS-2 in aiming to measure ASD symptoms. In comparison to a typically developing comparison group, van der Miesen et al (2018) found a GD sample had significantly higher scores on all subscales than a typically developing comparison group, but significantly lower than an ASD comparison group.

### Temporal precedence

None of the papers reviewed allows a robust assessment of the place of GD within the context of other MH problems (i.e., 'which came first?'). Becerra-Culqui et al [54] were able to show that there was a high degree of MH problems on records prior to first GD-related presentation, and other studies asked about psychiatric / MH history [30–33,35,37,38,61], but none reported prospective concurrent measurement of GD and other MH problems.

### Quality assessment

The CCAT quality ratings ranged from 45% to 96%, with a mean of 81%. All but one paper achieved an overall rating of 4 (good) or 5 (very good), with strengths and weaknesses within certain discrete categories; most papers achieved good ratings in the 'preliminaries' and 'introduction' categories, whereas the 'ethics' and 'discussion' categories were most likely to include lower ratings: 15 papers in each category achieved ratings below 4 (8 of these in both categories). Only one paper was rated as 3 (moderate quality) overall: Cohen-Kettenis & Van Goozen (2002) obtained low ratings across most categories, due to unclear sampling and diagnostic information, lack of information to permit replication, and conclusions which are not supported by the findings. Of the remainder, 14 were rated as high quality (4), and 17 as very high quality (5); see Table 4). There was no relationship between the year of publication and quality rating ($r$ = 0·3).

### Discussion

This systematic review synthesises the current evidence regarding the nature and prevalence of mental health (MH) problems in adolescents presenting for assessment for gender dysphoria (GD). We identified 32 papers published primarily in the past ten years, showing that MH problems, especially mood disorders / symptoms, are relatively common in the adolescent GD population. The few papers that drew on large samples or compared with normative data or non-GD comparison samples found a marked difference in prevalence rates of psychiatric diagnoses or assessment scores within an established clinical range. Comorbidity with neuro-developmental conditions, especially ASD, was also noted.

**Table 4. Mental health and cognitive status associated with GD.**

| ID | reference | design | Total N (% NM) | mental health measures | Findings | Notes |
|---|---|---|---|---|---|---|
| 19 | Akgul, et al. (2018) | obs & comp, prosp, x-sect | GD: 25 (52); control: 50 | Social Responsiveness Scale (SRS) | GD grp scored higher on total score ($p < .001$): GD grp mean 70.36 (16.7) Control mean 49.78 (16.9) 68% GD / 22% control in clinical range ($\chi^2 = 15.074$, $p < .001$) GD group NF higher than NM on SRS social ($p = 0.015$) and SRS ADHD-like subscales ($p = 0.019$). | Autism, psychosis, intellectual disability = exclusion criteria. Not clear if $p$ adjusted for multiple comparisons. |
| | | | | Behavior Rating Inventory of Executive Function (BRIEF) | GD grp scored higher on all 3 domains (all $p < .001$): Metacognitive GD 53.68 (9.8) / control 40.72 (7.5) Behavioral regulation GD 81.52 (12.9) / control 60.78 (11.6) Global executive composite GD 143.36 (21.8) / control 107.4 (19.34) | No clinical thresholds reported. Not clear if $p$ adjusted for multiple comparisons. |
| 25 | Becerra-Culqui, et al. (2018) | obs & comp, retro, x-sect | GD: 1082 (39.5) Comp: 21,317 | ICD-9 codes for MH diagnoses. Recorded as 'ever' and 'within 6 months of index date'. | 71% NM & 74% NF had MH diagnosis 'ever before index date'. 59% NM/ 65% NF had MH diagnosis in '6m before index date'. Most prevalent: • depressive disorders (49% NM/ 62% NF); • anxiety disorders (37% NM/ 39% NF); • attention deficit disorders (25% NM/ 16% NF). • Authors note selected findings: • particularly increased prevalence psychoses compared to reference females (NM: PR 101; 95% CI 14–4375; NF: PR 30; 95% CI 12–94). • Particularly elevated prevalence ASDs in NM (PR 261; 95% CI 43–10 734) & schizophrenia spectrum disorders in NF (PR 50; 95% CI 11–470) compared to ref females. • Particularly high prevalence of suicidal ideation (NM: PR 54; 95% CI 18–218; NF: PR 45; 95% CI 23–97) & self-harm (NM: PR 70 95% CI 9.0–159; NF: PR 144; 95% CI 14–4338) compared to reference males. Findings remain similar when included MH conditions recorded during hospitalisation. | Identification of sample based on health service use. Only large population study in this review. |
| 10 | Becker-Hebly et al (2020) | obs, prosp, longit, interv | 54 (15) (non-treatment group excluded as upper age range 18.01) | YSR | All scores differ significantly (i.e., poorer MH) from German norm. Baseline puberty suppression group (n = 11): 62.27 (8.96); 95% CI 56.26–68.29 Baseline GA hormones group (n = 32): 61.56 (9.17); 95% CI 58.26–64.87 Baseline GA surgery group (n = 11): 62.18 (8.78); 95% CI 56.28–68.08 | Severe psychiatric problems were an exclusion criterion. Compared to published YSR norms. |
| | | | | CGAS | Baseline puberty suppression group (n = 11): 67.27 (11.91); 95% CI 59.27–75.27 Baseline GA hormones group (n = 32): 73.13 (10.91); 95% CI 69.19–77.06 Baseline GA surgery group (n = 11): 66.36 (14.33); 95% CI 56.73–75.99 | No norm reference CGAS scoring criteria: 61–70 = 'some problems' 71–80 = 'doing all right' |
| | | | | Kidscreen-27 (mental dimension reported here) | All scores differ significantly (i.e., poorer MH) from German norm. Baseline puberty suppression group (n = 11): 39.04 (9.25); 95% CI 32.82–45.25 Baseline GA hormones group (n = 32): 36.16 (6.78); 95% CI: 33.72–38.60 Baseline GA surgery group (n = 11): 37.88 (6.53); 95% CI 33.49–42.27 | Compared to published Kidscreen-27 norms. |
| 26 | Chen, et al. (2016) | obs, retro, x-sect | 38 (42.1) | Concurrent psychiatric diagnoses | 24 (63.1%) had concurrent psychiatric and / or developmental diagnosis: • Depression 32%; • ADHD 16%; • ASD 13%; • history of suicidality / self-harm 13%. • 14 (36.8%) were on psychotropic medications. | |

(*Continued*)

**Table 4.** (*Continued*)

| ID | reference | design | Total N (% NM) | mental health measures | Findings | Notes |
|---|---|---|---|---|---|---|
| 4 | Chiniara, et al. (2018) | obs, retro, x-sect | 203 (23.2) | Mental health history–presence of psychiatric co-morbidities | 37% depressive disorder, 28% anxiety disorder, 33% suicidal thoughts, 30% self-harm. 34% prescribed medication for mood disorder (past or present).ASD & ADHD reported by natal sex:<br>• ASD 5% NF, 6% NM<br>• ADHD 15% NM, 0% NF ($p < .001$)<br>No other stat signif differences by natal sex. | Self-reported MH history |
| | | | | BDI-II | Severe depression 42% NF & 34% NM.<br>Positive suicide risk 10% NF, 10% NM. | |
| | | | | MASC2 | Significant anxiety 44% NF & 30% NM ($p$ .02). | |
| 15 | Cohen-Kettenis & Van Goozen (2002) | obs, retro, x-sect | CBCL: 29 (37.9)<br>DISC: 21 (52.4) | WISC-R / WAIS | CBCL sample: NM 99.4±14.3 / NF 104.9±14.5.<br>DISC-IV sample: NM 98.8±12.5 / NF 104.3±11.9. | impossible to discern overlap between the 29 (CBCL) and 21 (DISC) samples. |
| | | | | CBCL | 9/29 (6 NF) clinical range Total problem score. | |
| | | | | DISC IV | 1/11 in treatment had specific phobia (elevators) (NF), 1/11 transient tic disorder (NM). 1/10 not yet in treatment had ODD (NF), 1/10 overanxious disorder (NM). | |
| 20 | Costa, et al. (2015) | obs & comp, retro, longit, interv | 201 (37.8) (Comp grp: 169 assessed CAMHS Stockholm) | CGAS | Baseline scores significantly poorer in GD group<br>GD: 57.7±12<br>('Some noticeable problems–in more than one area')<br>Comp: 67.1 ± 12<br>('Some difficulty in a single area, but generally functioning pretty well')<br>Baseline scores significantly poorer in NM (55.4±12.7) than NF (59.2±11.8) ($p = 0.03$) | Part of 4 time-point follow-up study (see paper 3) |
| 13 | de Graaf et al. (2018). | obs & comp, retro, x-sect | (a) 252 (46.0)<br>(b) 71 (33.8)<br>(c) 26 (35.7)<br>(d) 610 (42.3) | CBCL | (a) Netherlands: Total mean 44.25±27.62 (38.8% clinical range)<br>(b) Belgium: Total mean 48.86±28.56 (54.3% clinical range)<br>(c) Switzerland: Total mean 52.96±23.73 (37.5% clinical range)<br> (d) UK: Total mean 51.24±29.78 (51.7% clinical range)<br> Netherlands significantly lower than UK on Total score ($p<0.01$); lower than UK and Swiss samples on Internalising problems (both $p<0.01$). No effects of natal sex.<br> Main effect between Netherlands and UK ($p<0.01$) on Total score; higher percentage of UK ($p<0.01$) and Swiss ($p<0.05$) sample in clinical range than Netherlands sample on Internalising score.<br> Signif more NF (53%) scoring in clinical range than NM (39%) ($p<0.01$) on Total score. Greater proportion of NF in clinical range than NM on Externalising problems ($p<0.01$). | Clinical range given as >90th percentile |
| | | | | YSR | (a) Netherlands: Total mean 45.67±22.05 (20.5% clinical range)<br>(b) Belgium: Total mean 60.49±23.91 (43.2% clinical range)<br>(c) Switzerland: Total mean 54.68±18.39 (15.4% clinical range)<br>(d) UK: Total mean 67.14±29.95 (46.4% clinical range)<br>Netherlands scored significantly lower than Belgium and UK (both $p<0.01$) on Total score; Swiss scored significantly lower than UK ($p<0.05$). No effect of natal sex.<br>Netherlands scored significantly lower than all other clinics on Internalising problems ($p<0.01$). No effect of natal sex.<br>Netherlands scored significantly lower than UK on Externalising problems ($p<0.01$). NF scored higher on Externalising problems than NM ($p<0.05$).<br> Significantly fewer in Netherlands in clinical range compared to Belgium ($p<0.01$) and UK ($p<0.01$) on Total score; signif fewer in Swiss sample in clinical range compared to UK ($p<0.01$) and Belgium ($p<0.05$). No effect of natal sex. | |

(*Continued*)

**Table 4.** (*Continued*)

| ID | reference | design | Total N (% NM) | mental health measures | Findings | Notes |
|---|---|---|---|---|---|---|
| | | | | YSR (continued) | Significantly fewer in Netherlands in clinical range compared to all other clinics (all $p<0.01$) on Internalising problems. More NM in clinical range than NF on Internalising problems ($p<0.01$). Higher percentage of UK sample in clinical range than Netherlands on Externalising score ($p<0.01$). No effect of natal sex. | |
| 16 | de Vries et al (2011a). Psychiatric comorbid. | obs, prosp, x-sect | 105 (50.5) | DISC-IV | DISC: prevalence 21% any anxiety disorder; 12.4% any mood disorder; 11.4% any disruptive behaviour disorder; 1.0% any substance use. 32.4% 1 or more DSM-IV diagnosis 15.2% 2 or more DSM-IV diagnoses 2.9% 3 or more DSM-IV diagnoses NM more likely to have 2 or more diagnoses (22.2% vs 7.7%; $p = 0.03$), mood disorder (20.8% vs 3.8%; $p = 0.008$), or social anxiety disorder (15.1% vs 3.8%; $p = 0.05$) than NF. | CBCL included in list of measures, but outcome not reported in this paper (see study 17 below) |
| 17 | de Vries, et al. (2011b). Puberty suppression | obs & comp, prosp, longit | 70 (47.1) | CBCL YSR CGAS BDI-II STPI (TPI & STAI) | CBCL: total score T0 60.7±12.76; 44% scoring in clinical range (other percentages not reported) YSR: total T score T0 55.46±11.56; 30% scoring in clinical range on internalising scale (other percentages not reported) CGAS (n41) T0 70.24±10.12 (Some problems–in one area only) BDI-II (n41) T0 8.31±7.12 (minimal range) TPI: Trait anger (n41) T0 18.29±5.54 STAI: Trait anxiety (n41) T0 39.43±10.07 | Part of longer follow-up study (see paper 3) Some gender differences at Baseline–hard to unpick as paper focused on comparisons over time. CBCL clinical range given as T>63. |
| 14 | de Vries, et al. (2016) | obs, comp, x-sect | (a) 139 (56.8) (b) 177 (53.1) | CBCL | a) (NL) 55.4% in clinical range for total problem score (Dutch manual) Normative non-referred: 9% b) (CA) 77.5% in clinical range for total problem score (US manual) Normative non-referred: 10% | Clinical range given as >90th percentile. In both samples, scores more likely to be in clinical range in internalising (rather than externalising) subscale and in boys rather than girls. The Toronto sample generally showed more problems than the Dutch sample. |
| | | | | YSR | 1. (NL) 40.6% in clinical range for total problem score (Dutch manual) Normative non-referred: 9% boys / 8% girls 2. (CA) 39.9% in clinical range for total problem score (US manual) Normative non-referred: 9% boys / 12% girls | |
| 27 | Edwards-Leeper, et al. (2017) | obs, retro, x-sect | 56 (53.6) | CBCL YSR | CBCL and YSR: all mean scores within normal range except CBCL internalising problems (borderline clinical: M = 60.21 ±12.44). 28.8% (CBCL) and 31.8% (YSR) scored within Total problems clinical range. 38.6% (YSR) and 48.1% (CBCL) scored within internalising problems clinical range. Self-harm reported by 4/52 CBCL & 4/45 YSR; suicidal ideation reported by 6/51 CBCL & 11/46 YSR | some evidence of MH declining with age. |
| | | | | CDI | mean scores all in average range | |
| | | | | RCMAS | mean scores all in average range | |
| | | | | Piers-Harris 2 | mean scores all in average range | |
| 5 | Feder, et al. (2017) | obs, retro, x-sect | 97 (38.1) | clinical interview determined DSM-5 status re GD and eating disorder | 5/97 concurrent ED diagnosis—3 anorexia restrictive, 1 anorexia binge/purge, 1 Avoidant Restrictive Food Intake Disorder (ARFID). 10/97 ED symptoms but no ED assessment. 12/15 with ED / ED symptoms were NF. | |

(*Continued*)

**Table 4.** (Continued)

| ID | reference | design | Total N (% NM) | mental health measures | Findings | Notes |
|---|---|---|---|---|---|---|
| 12 | Fisher, et al. (2017) | obs & comp, prosp, x-sect | 46 (43.5) control: 46 (39.1) non-referred–NR | YSR (Italian) | GD scored sig higher than NR on total (60.91 ± 7.46 vs. 55.30 ± 6.16; p < 0.0001) and internalising (62.43 ± 11.18 vs. 53.57 ± 11.64; p = 0.001) (not externalising). GD had more in clinical range than NR on total (17.4 vs. 10.9; p < 0.0001) and internalising (47.8 vs. 17.4%; p < 0.0001) (not externalising). | |
| | | | | MAST | GD sig higher than NR on 'attraction to death' (2.98 ± 0.57 vs. 2.17 ± 0.58; p < 0.0001) and 'repulsion by life' (3.04 ± 0.46 vs. 2.08 ± 0.56;p < 0.0001), lower scores on 'attraction to life' (3.32 ± 0.55 vs. 4.05 ± 0.49; p < 0.0001). | |
| | | | | Body Uneasiness Test (BUT) | GD sig higher than NR on body uneasiness (Global Severity Index) (3.05 ± 0.49 vs. 0.61 ± 0.58; p < 0.0001). | |
| 6 | Heard, et al. (2018) | obs, retro, x-sect | 174 (29.9) (also 25/96 FU survey Aug 2015-Jul 2016, not reported here) | any mental health diagnosis (from clinical records—no measure given) | 66 (38%) had pre-existing or current MH diagnosis, most commonly anxiety or depression (n43, 30.5%). Self-harm in 36 (21%) ASD / ASD traits in 4 / 2 respectively | Focus of paper was the survey, so little discussion of these results. |
| 21 | Holt et al. (2016) | obs, retro, x-sect | 177 (37.2) | 'associated difficulties' as recorded in notes | Most common: low mood/depression (42%), and self-harming (39%). Self-harm most common in NF; low mood/depression most common in NM. NF reported signif more self-harm than NM (p<0.01). ASC 18.5% NM / 10.2% NF (p<0.01) ADHD 12.3% NM / 5.8% NF Anxiety symptoms 21.0% NM / 23.4% NF Eating difficulties 12.3% NM / 13.9% NF Self-harm (p < .001), suicide attempts (p < .05), low mood/ depression (p < .001), eating difficulties (p < .01) increased with age of sample (but note youngest still in childhood). | |
| 8 | Kaltiala-Heino et al. (2015) | obs, retro, x-sect | 47 (12.8) | previous and current psychiatric history | 35/47 (75%) had been previously or currently attending child and adol psychiatry services for reasons other than GD. • 64% depression • 55% anxiety • 53% suicidal thoughts / self-harm • 26% autism • 13% psychotic symptoms • 11% ADHD • 9% conduct disorders • 4% substance misuse | Qualitative analysis identified 5 distinct groups based on age of onset and psychiatric comorbidity. |
| 9 | Kaltiala-Heino et al (2019) | obs, retro, x-sect | 99 (15.2) | current or previous specialist-level psychiatric treatment history (yes/no) | 76/99 (77%) had been previously or currently attending child and adol psychiatry services for reasons other than GD. • 66% depression • 58% anxiety • 54% suicidality / self-harm • 17% autism • 12% psychotic symptoms • 12% conduct disorders • 7% substance misuse • 8% ADHD | Data reported by natal sex in paper but combined here as no significant differences. Likely significant overlap with paper above. |
| 28 | Kuper, et al. (2019a)– Exploring the developmental histories. . . | obs, retro, x-sect | 224 (39.4) | Depression (QIDS), Anxiety (SCARED) Bodily dissatisfaction (BIS) | QIDS: mean 9.5±5.2; 24.6% / 16.8% in moderate / severe range. SCARED: mean 30.8±15.8; 60.9% clinically elevated range. BIS: mean 68.6±16.0. No norms; absolute possible range = 30–150. Dissatisfaction with figure and voice associated with depression (both p < .01). No comparison by natal sex (see Kuper et al (2020) below). | QIDS & SCARED age ≥9 only BIS age ≥12 only Subgroup Ns not reported. Same as sample below (Kuper et al 2020). |

(*Continued*)

**Table 4.** (*Continued*)

| ID | reference | design | Total N (% NM) | mental health measures | Findings | Notes |
|---|---|---|---|---|---|---|
| 29 | Kuper, et al. (2019b)– Baseline mental health... | obs, prosp, x-sect | 149 (35.6) | YSR / CBCL BIS | YSR/CBCL taken together, percentage range clinically elevated:<br>• 34.5%–50.3% Total Problems,<br>• 42.7%–62.8% Internalizing Problems,<br>• 42.3%–58.3% total Competency scales.<br>• 30.0–37.8% depression,<br>• 20.0–32.9% anxiety,<br>• 15.5–33.6% obsessive compulsive,<br>• 15.5–30.1% post-traumatic stress,<br>• 2.7–10.5% AD/H problems.<br>All rest below 10%.<br>NF scored significantly higher on YSR Total and YSR Internalising; also on depressive, anxiety, social, obsessive-compulsive, and post-traumatic stress problems scales (all $p<0.01$).<br>BIS: mean 69.6±15.4. No norms; absolute possible range = 30–150. No difference by gender. | Clinical range given as T>63. |
| 30 | Kuper et al (2020) | obs, prosp, longit, interv | 148 (37) | BIS QIDS SCARED | BIS: mean 69.9 (15.6) R 0–116<br>QIDS (self): mean 9.4 (5.2) R 0–27. Within 'mild' range.<br>QIDS (clin): mean 5.8 (4.2) R 0–27. Within 'mild' range.<br>SCARED (total): mean 32.4 (16.3) R 0–82<br>NF significantly higher clinician-reported and self-reported depressive symptoms, total anxiety, panic symptoms, and school avoidance symptoms. All effect sizes small, however. | Same sample as Kuper et al (2019a) (and presumably 2019b?), with follow-up data. |
| 11 | Levitan, et al. (2019) | obs, prosp, x-sect | 180 (18.9) | YSR | Internalising: T scores 65.2±10.64, 55% within clinical range<br>Externalising: 55.5±7.73, (% within clinical range not given as not stat signif diff from population norm)<br>Total: 62.6±8.79, 44% within clinical range. | Predictive model found NF and higher age signif predictor of Total problem scores. High GFF score also signif predictor of Total problem score. |
| | | | | General family functioning (GFF) from MFAD | Mean: 1.97±0.59; 34% above threshold for 'problematic' GFF | |
| 1 | Mahfouda, Panos et al (2019) | obs, retro, x-sect | 104 (24) | Psychiatric history (parent) | 77.90% (n = 81) reported history any MH problems, deliberate self-harming behaviours and/or suicidal intent or attempts<br>9.62% (n = 10) reported formal ASD diagnosis; (information available 10.58% (n = 11 of the sample)) | Other subscales reported |
| | | | | SRS-2 (parent) | Total: mean 60.68 (14.06), within 'mild' range<br>Using DSM-5 subscale:<br>17.3% mild<br>9.6% moderate<br>22.1% severe (indicating ASD)<br>No differences by gender. | |
| | | | | YSR | Total t-score: 64.08 (10.94)<br>54.8% in clinical range<br>Odds of falling into clinical range for YSR Total higher for those with indicated ASD on SRS-2 ($p = 0.003$). No diff by Int or Ext score. | |
| | | | | PedsQL | PedsQL (psych): 55.95 (19.77)<br>Those with indicated ASD significantly lower (45.25±19.52) than those without (58.79±19.33) | |

(*Continued*)

**Table 4.** (Continued)

| ID | reference | design | Total N (% NM) | mental health measures | Findings | Notes |
|---|---|---|---|---|---|---|
| 22 | Matthews, et al. (2019) | obs, retro, x-sect | 185 (52.4) | CGAS | CGAS:<br>• looked-after (n5) mean 48.6±4.08 'moderate impairment in most areas or severe in one area'<br>• adopted (n5) mean 58.20±12.87 'Variable functioning with sporadic difficulties or symptoms in several but not all social areas'<br>• YPBF (n127) mean 61.34±13.72 'Some difficulty in a single area, but generally functioning pretty well'<br>Comorbidity with other diagnoses:<br>• looked-after 55.6%<br>• adopted 28.6%<br>• YPBF 41.4%<br>No breakdown by diagnostic category. | Data tables in paper were mislabelled |
| 31 | Moyer, et al. (2019) | obs, retro, x-sect | 194 (24.2) | PHQ-9 | At initial consultation, 48.1% clinically significant depression symptoms; 35.9% thoughts of death / self-harm.<br>Gender non-binary more likely to endorse thoughts of death / self-harm ($p<0.05$) than those affirming binary gender identity | Whole sample included follow-ups. Initial consultation data reported here (n = 79) |
| | | | | GAD-7 | GAD-7: 73.7% clinically significant anxiety symptoms. | |
| 32 | Nahata, et al. (2017) | obs, retro, x-sect | 79 (35.4) | Mental health: diagnoses, self-harm | Diagnoses:<br>• Depression 78.5%,<br>• Anxiety 63.3%,<br>• PTSD 22.8%,<br>• eating disorders 12.7%,<br>• ASD 6.3%,<br>• bipolar disorder 5.1%.<br>Suicidal ideation 74.7%, self-harm 55.7%, one or more suicide attempts 30.4%.<br>No significant differences by gender. | |
| 23 | Russell et al (2020) | obs, retro, x-sect | 95 (40) | SRS-2 | SRS-2 (total): 56.66 (2.29) NM (within normal range)<br>62.46 (1.87) NF (within mild clinical range)<br>Non-sig diff | Includes FU one year later, baseline only reported here |
| 24 | Skagerberg et al (2013) | obs, x-sect | 141 (40) | YSR | Mean Total problems score 62.5±10.9 and Externalising score 56.47±11.17 in normal range (t<60). Internalising score 63.23 ±12.35 in clinical range (t>63). NM signif higher than NF on Internalising score ($p<0.05$).<br>NM: 47% clinical, 16% borderline (t = 60–63) on Total score<br>NF: 49% clinical, 17% borderline on Total score | |
| 7 | Sorbara et al (2020) | obs & comp, retro, x-sect | 300 (23.7) | Youth / caregiver report formal diagnoses of depressive, anxiety, ASD. Active psychoactive medication use, suicidal ideation, history self-harm / suicide attempt | 18 (6%) reported ASD<br>78% reported ≥1 mental health problem<br>40.0% depressive disorder<br>44.3% anxiety disorder<br>34.7% self-harm; 12.3% current suicidal ideation; 47.3% considered suicide, 14.0% attempted suicide<br>31.3% taking psychoactive medication<br>Those ≥15 yrs signif higher rates depressive disorders, self-harm, previous suicidal ideation, suicide attempts, and psychoactive medication use (all p<0.05).<br>NF OR = 3.41 (95% CI: 1.42–8.19) self-harm compared to NM.<br>With each 1 year incr in age, odds of psychoactive medication use incr by 1.31 (95% CI: 1.05–1.63). | |
| 2 | Tack, et al. (2016) | obs, retro, longit, interv | 45 (0) | Psychiatric morbidity at baseline | 11/43 (25.6%) psychiatric comorbidity.<br>No break down by diagnostic category. | |

(*Continued*)

**Table 4.** (Continued)

| ID | reference | design | Total N (% NM) | mental health measures | Findings | Notes |
|---|---|---|---|---|---|---|
| 3 | Tack, et al. (2017) | obs, retro, longit, interv | 27 (100) | Psychiatric morbidity at baseline | 6/27 (22.2%) psychiatric comorbidity (ASD = 3, depression = 1, ADHD = 1, ASD + depression = 1). | Reported as preceding GD diagnosis, likely due to this being reported at intake (so by definition prior to GD dx). |
| 18 | van der Miesen, et al. (2018) | obs & comp, prosp, x-sect | 490 (50.6) (Comp grp 1: 2507 from Dutch schools. Comp grp 2: 196 with ASD dx Dutch psychiatry clinic) | Children's Social Behaviour Questionnaire (CBSQ) | GD grp total: 20.58±15.71 TD grp total: 11.69±11.49 ASD grp total 42.08±16.72 (≥38 indicative of ASD) GD grp higher scores all subscales and total than TD grp (but lower than ASD grp) (p < 0.001). No effect of gender on Total score | A lot of analyses by subscale–is this relevant? |

**Key:**

NM / NF: Natal Male / Natal Female.

DSM-5: Diagnostic and Statistical Manual of Mental Disorders, 5th Edition; ICD-9: International Classification of Diseases, Ninth revision.

BDI-II: Beck Depression Inventory; BIS: Body Image Scale; BUT: Body Uneasiness Test; CBCL / YSR: Child Behavior Checklist / Youth Self Report; CBSQ: Children's Social Behaviour Questionnaire; CDI: Children's Depression Inventory; CGAS: Children's Global Assessment Scale; GAD-7: General Anxiety Disorder; GFF: General Family Functioning (derived from MFAD); MASC2: Multidimensional Anxiety Scale for Children, 2nd Edition; MAST: Multi-Attitude Suicide Tendency Scale; MFAD: McMaster's Family Assessment Device; PHQ-9: Patient Health Questionnaire; Piers-Harris 2: Piers-Harris Children's Self-Concept Scale; STPI: QIDS: Quick Inventory of Depressive Symptoms; RCMAS: Revised Children's Manifest Anxiety Scale; SCARED: Screen for Childhood Anxiety Related Emotional Disorders; STAI: Spielberger's Trait Anxiety Scale; STPI: State-Trait Personality Inventory; TPI: Speilberger's Trait Anger Scale.

ADHD: Attention Deficit / Hyperactivity Disorder; ASD: Autism Spectrum Disorder; GD: Gender Dysphoria / Dysphoric; PTSD: Post Traumatic Stress Disorder.

TD: Typically Developing; T0, T1, etc: Time 0, Time 1, etc (referring to point of data collection in longitudinal studies).

YPBF: young people living with their birth family.

Whilst the overall picture is one of increased levels of MH problems compared to the general population, the prevalence and nature of MH problems varied across studies. Some differences were noted by geographic region, for example, with those from the Netherlands generally scoring more positively than those in other regions, and the UK and Finland faring poorly [37,42,43]. There are a range of likely reasons for this, the most usually cited being that the culture in the Netherlands is much more open and supportive regarding gender fluidity and there is simply less stigma [42,43]. Other potential reasons, such as inequitable access to appropriate specialist services, have been ruled out by some authors as the countries included in comparison studies (Netherlands, UK, Canada) have universal health care [43]. However, it would be useful to look at factors such as waiting times to receiving specialist care: one paper mentioned similar waiting times in the Netherlands and the UK [58] based on information available on the centre websites, but this has not been systematically investigated and reported and it would seem that, at least in the UK, waiting times have recently increased considerably due to greater demand [81]. In other regions, initial assessment may take place in non-specialist services, which may be more accessible and therefore may increase the likelihood of being seen before considerable mental distress develops.

Stage of presentation for treatment is also relevant and varied across studies. The point at which young people are being assessed for admission to intervention services is likely to be an acutely stressful time; some studies found that adolescents with poorer mental wellbeing were those presenting at a more advanced stage of puberty [54], and in systems / regions where

waiting times are necessarily longer, it is likely that secondary sexual characteristics have already begun to develop, and distress increased.

Another distinguishing factor between regions is the underlying level of mental wellbeing in the population. Reports show that adolescents in some regions, such as the Nordic countries and the Netherlands, show better levels of mental wellbeing than those in the UK [82,83], for example. There is acknowledged inequality in service provision and accessibility in the UK, and in general thresholds have gone up so that young people are waiting months to years for initial assessment for a mental health concern (by which point problems have become severe) [84], and this will apply to other regions. It is likely, therefore, that some young people presenting for GD intervention have already experienced years of untreated psychological distress which may or may not have preceded their GD. In the US, there is inherent inequality and significant inter-state variation in the health system, and insurance denial has been noted as a barrier to timely treatment [14,61]. All of these contextual factors, and more, need to be properly characterised and considered in describing adolescent GD populations.

Some differences were also noted between sexes, but the findings were not consistent. In some papers, NF participants scored more poorly than NM [33,40,48], but the reverse was true in others, and some studies found no differences by natal sex. In some studies, the usual pattern of NF scoring more highly on internalising problems and NM more highly on externalising was reversed, suggesting psychopathology in these young people aligned more with their identified gender than their assigned gender [42,43,53]. But again, this pattern was not consistently observed (e.g., Kuper et al, 2019b [58]).

The findings in many studies that NF have poorer mental wellbeing, along with the very rapid increase in NFs presenting for treatment (see paper 1, ref), is notable and requires careful monitoring. The aetiology of GD is not fully understood, and the implications of this demographic change are important. Most papers attribute the increase in young people presenting for treatment to cultural shifts in acceptance of gender fluidity and greater availability of services. Whilst these factors are no doubt important, this alone probably does not explain the dramatic increase in NF presentations: there remains the possibility, not apparently explored in this literature, that modern sociocultural pressures associated with womanhood / femininity are influencing this generation's propensity to seek treatment.

Despite the relatively large number of papers in recent years on this subject, there are questions which it is not possible to answer with the given methods. Most papers were cross-sectional in nature, and those that were longitudinal tended to follow participants through their treatment journey (which will be the focus of paper 3). Samples were often relatively small, and there is an acknowledged lack of representation, with little if any racial diversity among study participants, and socioeconomic status rarely measured / presented. It is also noteworthy that almost all the participants in the included studies are characterised as gender binary: other identities remain under-represented. Diversity in methods of measuring and reporting mental health status limits the ability to synthesise findings and draw over-arching conclusions.

The reality is that young people presenting for GD intervention do so within a highly complex context which studies need to better characterise if we are to fully understand the underlying drivers and improve responses to GD and MH comorbidity. We understand very little about the development of MH problems prior to presentation at GD services, and so do not have a clear understanding of the place of GD within the broader context of young people's MH. Simple developmental screening throughout childhood, such as that in place in some Nordic countries, and the ability to link clinical records to population datasets, would allow us to more clearly address issues of temporal precedence and complex interaction of contextual variables. This may be especially true in relation to the co-occurrence of neurodevelopmental disorders. Whilst studies have found an increased representation of autistic features and ASD

diagnoses within the adolescent GD population, the interplay of these characteristics cannot be explored without better longitudinal, population-based studies. Indeed, problems in family functioning, peer relationships, and social relationships more broadly were noted in several of the included studies in the present sample [40,42,43]. To what extent GD may be a reaction to a range of preceding psychosocial stressors for a sub-sample of those presenting for services needs to be understood.

Some studies were keen to present an optimistic picture [56], and it is worth noting that in many studies the majority of participants were not experiencing MH problems. Most of the literature in this review comes from a medical standpoint, which tends towards a deficit model of characterisation as the focus is on identifying and treating illness. Some qualitative studies have addressed more positive accounts and the importance of taking a non-pathologising stance [85], but this research tends to focus on adults' experiences. None in the present sample focused on characterising resilience factors, and this will be an important direction for future research.

## Strengths and limitations

This review has strength in the broad search strategy and thorough hand screening process applied. There are methodological limitations which need to be considered. The broad initial search criteria led to the need for some narrowing of criteria following initial screening (but prior to full-text screening). The addition of parameters regarding type of publication, upper age of participants, and the clinical verification of GD naturally narrowed the pool of papers and therefore may have meant papers with important findings have been excluded (for example, if a paper included an upper age limit of 21 even though the majority were younger than 18). We endeavoured to record all papers that only narrowly missed inclusion on the age criterion (Table 5), but literature that was excluded on the basis of type (i.e., conference proceedings and grey literature) were not included at this stage and so the potential contribution of this body of work cannot be quantified or assessed. Similarly, we excluded papers only reporting qualitative findings as these would not have been able to provide the type of data needed for our research questions, but we acknowledge the potential loss of richer lived experience information in so doing. We opted to use a quality assessment tool for studies of diverse designs (CCAT). This allowed all papers to be rated using the same system, but also involved reviewers having to make subjective ratings rather than apply a strictly quantifiable checklist. This may have led to issues with quality, such as over-statement of the significance of findings, not being sufficiently prominent.

Although we were able to include 32 papers from a range of countries in this review, over a third arose from two well-established treatment centres: those in Amsterdam and London. The Amsterdam team has led the way in developing assessment and treatment protocols for GD and provides a wealth of data over a long period (since 1996 within the included papers), and the London GIDS is a hub for the whole of the United Kingdom now dealing with hundreds of referrals per year. This presents the advantage of being able to observe the adolescent GD population over a long period of time, assessed using the same or similar tools, and within a relatively stable social context. It is not clear, however, what proportion of young people experiencing GD have access to these national specialist centres and how many may be accessing private facilities or self-medicating with hormones obtained via other routes: we do not know how representative these samples are. Another disadvantage is that most of the papers included in this review are likely to include data from the same samples of participants, also limiting generalisability. The overlap between samples was rarely overtly stated, and there is a risk that readers may add greater weight to collective findings than is warranted. There is a

**Table 5. Papers excluded at second full text screen (i.e., closely missed meeting inclusion criteria).**

| Reference | Date | Location & setting | Reason for exclusion | Notes |
|---|---|---|---|---|
| Achille, C., Taggart, T., Eaton, N. R., Osipoff, J., Tafuri, K., Lane, A., & Wilson, T. A. (2020). Longitudinal impact of gender-affirming endocrine intervention on the mental health and well-being of transgender youths: preliminary results. International Journal of Pediatric Endocrinology, 2020, 8. | 2020 | USA New York, Stoney Brook Children's Hospital | 1 Mean age 16.2±2.2 | 66% NF MH improved with endocrine intervention Small sample (n = 50) |
| Aitken, et al (2015). Evidence for an altered sex ratio in clinic-referred adolescents with gender dysphoria. Journal of Sexual Medicine, 12(3), 756–763. | 2015 | 1) CANADA Gender Identity Service, Child, Youth and Family Services (CYFS), Toronto 2) NETHERLANDS Center of Expertise on Gender Dysphoria (CEGD) | 1, 2 1) unclear if GD clinically verified 2) Eldest participant 19 years | Useful information in change of gender in those presenting to services over time. |
| Alastanos, J. N., & Mullen, S. (2017). Psychiatric admission in adolescent transgender patients: A case series. The Mental Health Clinician, 7(4), 172–175. | 2017 | USA 'an inpatient psychiatry unit' | 3 Selected case series | All 5 participants were psychiatric inpatients within a 5 week period |
| Alberse, et al. (2019). Self-perception of transgender clinic referred gender diverse children and adolescents. Clinical Child Psychology and Psychiatry, 24(2), 388–401. | 2019 | NETHERLANDS Center of Expertise on Gender Dysphoria (CEGD), Amsterdam | 1 Max age 18.03 | Poor self-perception common. NF perceive themselves more positively in general. |
| Alexander, G. M., & Peterson, B. S. (2004). Testing the prenatal hormone hypothesis of tic-related disorders: gender identity and gender role behavior. Development & Psychopathology, 16(2), 407–420. | 2004 | USA Child Study Center of Yale University in New Haven, CT | 6 | Participants receiving treatment for Tourette Syndrome, not presenting for GD |
| Amir, H., Oren, A., Klochendler Frishman, E., Sapir, O., Shufaro, Y., Segev Becker, A.,. . . Ben-Haroush, A. (2020). Oocyte retrieval outcomes among adolescent transgender males. Journal of Assisted Reproduction & Genetics, 37(7), 1737–1744. | 2020 | ISRAEL IVF Unit, Fertility Institute in Tel Aviv Sourasky Medical Center and IVF and the Infertility Unit, Helen Schneider Hospital for Women, Rabin Medical Center | 3 | Sample was 11 NF presenting specifically for fertility preservation |
| Amir, H., Yaish, I., Oren, A., Groutz, A., Greenman, Y., & Azem, F. (2020). Fertility preservation rates among transgender women compared with transgender men receiving comprehensive fertility counselling. Reproductive Biomedicine Online, 41(3), 546–554. | 2020 | ISRAEL Gender Dysphoria Clinic at Dana-Dwek Children's Hospital, Gender Clinic, Tel Aviv Sourasky Medical Center | 0 | Included in Paper 1 (epidemiology) |
| Anzani, A., Panfilis, C., Scandurra, C., & Prunas, A. (2020). Personality Disorders and Personality Profiles in a Sample of Transgender Individuals Requesting Gender-Affirming Treatments. International Journal of Environmental Research & Public Health, 17(5), 27. | 2020 | ITALY gender clinic at Niguarda Ca' Granda Hospital, Milan | 3 | Very small sub-sample (n = 4) in adolescent age range |
| Arnoldussen, et al. (2019). Re-evaluation of the Dutch approach: are recently referred transgender youth different compared to earlier referrals? European Child and Adolescent Psychiatry. | 2019 | NETHERLANDS Center of Expertise on Gender Dysphoria (CEGD), Amsterdam | 1 Max age 18.08 | From 2000–2016, sharp increase in cases from 2012 to 2016; sharp uptick in NF relative to NM since 2013 |
| Avila, J. T., Golden, N. H., & Aye, T. (2019). Eating Disorder Screening in Transgender Youth. Journal of Adolescent Health, 65(6), 815–817. | 2019 | USA 'an academic multidisciplinary gender clinic' Stanford University School of Medicine* | 1, 2 No distinct data on adolescents in sample. | Most (63%) disclosed weight manipulation for gender-affirming purposes, including 11% of NF for menstrual suppression. |

*(Continued)*

**Table 5.** (Continued)

| Reference | Date | Location & setting | Reason for exclusion | Notes |
|---|---|---|---|---|
| Barnard, E. P., Dhar, C. P., Rothenberg, S. S., Menke, M. N., Witchel, S. F., Montano, G. T.,. . . Valli-Pulaski, H. (2019). Fertility Preservation Outcomes in Adolescent and Young Adult Feminizing Transgender Patients. Pediatrics, 144(3). | 2019 | USA Magee-Womens Research Institute, Pittsburgh* | 1, 3 | Very small subsample (n = 11); only 4 in adolescence at time of assessment (consultation). |
| Becerra-Fernández, A., Rodríguez-Molina, J., Ly-Pen, D., Asenjo-Araque, N., Lucio-Pérez, M., Cuchí-Alfaro, M.,. . . Aguilar-Vilas, M. V. (2017). Prevalence, Incidence, and Sex Ratio of Transsexualism in the Autonomous Region of Madrid (Spain) According to Healthcare Demand. Archives of Sexual Behavior, 46(5), 1307–1312. doi:10.1007/s10508-017-0955-z | 2017 | SPAIN Gender Identity Unit, Madrid | 1 No separate data on adolescents (includes ≥18 yrs) | Higher prevalence rate than other countries: attributed to easily accessible services and positive social and legal climate in Spain. |
| Bechard, M., VanderLaan, D. P., Wood, H., Wasserman, L., & Zucker, K. J. (2017). Psychosocial and Psychological Vulnerability in Adolescents with Gender Dysphoria: A "Proof of Principle" Study. Journal of Sex & Marital Therapy, 43(7), 678–688. | 2017 | CANADA Gender Identity Service, CYFS, Toronto | 1 No separate data on adolescents (includes ≥18 yrs) | Mean of 5.56/13 'psychological vulnerability factors' amongst sample. |
| Becker, I., Auer, M., Barkmann, C., Fuss, J., Moller, B., Nieder, T. O.,. . . Richter-Appelt, H. (2018). A Cross-Sectional Multicenter Study of Multidimensional Body Image in Adolescents and Adults with Gender Dysphoria Before and After Transition-Related Medical Interventions. Archives of Sexual Behavior, 47(8), 2335–2347. | 2018 | GERMANY Department of Child and Adolescent Psychiatry, Psychotherapy, and Psycho-Somatics, University Medical Centre, Hamburg | 1 No separate data on adolescents (includes ≥18 yrs) | Body image generally poor; some (but not all) aspects of poor body image improved with intervention. |
| Biggs, M. (2020). Gender Dysphoria and Psychological Functioning in Adolescents Treated with GnRHa: Comparing Dutch and English Prospective Studies. Archives of Sexual Behavior, 49(7), 2231–2236. | 2020 | Secondary data: NETHERLANDS and UK | 5 No original data | Useful critique of existing data: compares Dutch and English samples |
| Bonifacio, J. H., Maser, C., Stadelman, K., & Palmert, M. (2019). Management of gender dysphoria in adolescents in primary care. Cmaj, 191(3), E69-E75. | 2019 | N/A–review paper (Authors based in Toronto, CANADA) | 5 | Increase in cases will mean primary care services have to be prepared |
| Bradley, S. J. (1978). Gender identity problems of children and adolescents: The establishment of a special clinic. The Canadian Psychiatric Association Journal / La Revue de l'Association des psychiatres du Canada, 23(3), 175–183. | 1978 | CANADA Toronto, southwestern Ontario | 3 Small N. | No objective data–clinical impressions only. Interesting early paper on GD and Toronto clinical service. |
| Brocksmith, V. M., Alradadi, R. S., Chen, M., & Eugster, E. A. (2018). Baseline characteristics of gender dysphoric youth. Journal of Pediatric Endocrinology and Metabolism, 31(12), 1367–1369. | 2018 | USA pediatric endocrine clinic, Riley Hospital for Children, Indianapolis | 1 No separate data on adolescents (includes ≥18 yrs) | High proportion of NF to NM 50% overweight / obese Anxiety higher in NF Some indication of adolescent-onset being more common in recent (post-2014) participants |
| Bui, H. N., Schagen, S. E. E., Klink, D. T., Delemarre-Van De Waal, H. A., Blankenstein, M. A., & Heijboer, A. C. (2013). Salivary testosterone in female-To-male transgender adolescents during treatment with intra-muscular injectable testosterone esters. Steroids, 78(1), 91–95. | 2013 | NETHERLANDS VU Medical Center, Amsterdam | 1, 3 | Technical paper focused on novel method of measuring salivary testosterone levels. |

(Continued)

**Table 5.** (Continued)

| Reference | Date | Location & setting | Reason for exclusion | Notes |
|---|---|---|---|---|
| Burke, S. M., Kreukels, B. P. C., Cohen-Kettenis, P. T., Veltman, D. J., Klink, D. T., & Bakker, J. (2016). Male-typical visuospatial functioning in gynephilic girls with gender dysphoria—Organizational and activational effects of testosterone. Journal of Psychiatry and Neuroscience, 41 (6), 395–404. | 2016 | NETHERLANDS Center of Expertise on Gender Dysphoria (CEGD), Amsterdam | 3 | Small selected sample, unlikely to be representative. Psychiatric disorder an exclusion criterion. |
| Butler, G., De Graaf, N., Wren, B., & Carmichael, P. (2018). Assessment and support of children and adolescents with gender dysphoria. Archives of Disease in Childhood, 103(7), 631–636. | 2018 | UK Gender Identity Development Service, Tavistock, London | 5 Commissioned review | Some data on age and gender |
| Calzo, J. P., & Blashill, A. J. (2018). Child Sexual Orientation and Gender Identity in the Adolescent Brain Cognitive Development Cohort Study. JAMA Pediatrics, 172(11), 1090–1092. | 2018 | USA San Diego (Adolescent Brain Cognitive Development (ABCD) study) | 2, 4 | Self-identification of sexuality and gender status, with very small N identifying as transgender. Sample aged 9–10 yrs. |
| Chen, D., Simons, L., Johnson, E. K., Lockart, B. A., & Finlayson, C. (2017). Fertility Preservation for Transgender Adolescents. Journal of Adolescent Health, 61(1), 120–123. | 2017 | USA Gender & Sex Development Program (GSDP), Ann & Robert H. Lurie Children's Hospital of Chicago | 3 | Very small sample. 11/13 were adolescents. Sample was 11 adolescents presenting specifically for fertility preservation. |
| Chodzen, G., Hidalgo, M. A., Chen, D., & Garofalo, R. (2019). Minority Stress Factors Associated With Depression and Anxiety Among Transgender and Gender-Nonconforming Youth. Journal of Adolescent Health, 64(4), 467–471. | 2019 | USA Division of Adolescent Medicine, Ann & Robert H. Lurie Children's Hospital of Chicago* | 2 Sample self-identified as transgender and gender-nonconforming (TGNC) | High levels of anxiety and depression. |
| Clark, T. C., Lucassen, M. F. G., Bullen, P., Denny, S. J., Fleming, T. M., Robinson, E. M., & Rossen, F. V. (2014). The health and well-being of transgender high school students: Results from the New Zealand adolescent health survey (youth'12). Journal of Adolescent Health, 55(1), 93–99. | 2014 | NEW ZEALAND National population-based survey | 2 Survey data. Participants self-identify as transgender. | Adolescents identifying as transgender have considerable health and wellbeing needs relative to their peers. |
| Cohen, L., De Ruiter, C., Ringelberg, H., & Cohen-Kettenis, P. T. (1997). Psychological functioning of adolescent transsexuals: Personality and psychopathology. Journal of Clinical Psychology, 53(2), 187–196. | 1997 | NETHERLANDS VU Medical Center, Amsterdam | 1 | Mean age 17.2±1.81 Rorschach methodology–not useful for our review |
| Cohen-Kettenis, P. T., & Van Goozen, S. H. M. (1997). Sex reassignment of adolescent transsexuals: A follow-up study. Journal of the American Academy of Child and Adolescent Psychiatry, 36(2), 263–271. | 1997 | NETHERLANDS University Medical Centre, Utrecht (moved to VUMC / CEGD in 2002). | 1, 3 | GD resolved at post-surgery follow-up. No expression of regret. |
| Coolidge, F. L., Thede, L. L., & Young, S. E. (2002). The heritability of gender identity disorder in a child and adolescent twin sample. Behavior Genetics, 32(4), 251–257. | 2002 | USA Colorado Springs, Colorado | 2 | Twin methdology to investigate heritability. |
| Day, J. K., Fish, J. N., Perez-Brumer, A., Hatzenbuehler, M. L., & Russell, S. T. (2017). Transgender Youth Substance Use Disparities: Results From a Population-Based Sample. Journal of Adolescent Health, 61(6), 729–735. doi:10.1016/j.jadohealth.2017.06.024 | 2017 | USA 2013–2015 Biennial Statewide California Student Survey | 2 Survey data. Participants self-identify as transgender. | Transgender youth at increased risk for substance misuse. Some psychosocial factors may mediate this. |

(Continued)

**Table 5.** (Continued)

| Reference | Date | Location & setting | Reason for exclusion | Notes |
|---|---|---|---|---|
| de Graaf, et al. (2018). Sex ratio in children and adolescents referred to the Gender Identity Development Service in the UK (2009–2016). Archives of Sexual Behavior, 47(5), 1301–1304. doi:10.1007/s10508-018-1204-9 | 2018 | UK<br>Gender Identity Development Service, Tavistock, London | 2<br>Referrals only–no GD clinical verification | Significant increases year-on-year, from only 39 adolescent referrals in 2009 to almost 1500 in 2016; average increase rate of referrals higher in NF. |
| de Graaf, N. M., Carmichael, P., Steensma, T. D., & Zucker, K. J. (2018). Evidence for a Change in the Sex Ratio of Children Referred for Gender Dysphoria: Data From the Gender Identity Development Service in London (2000–2017). Journal of Sexual Medicine, 15(10), 1381–1383. | 2018 | UK<br>Gender Identity Development Service, Tavistock, London | 4 | NM referred at younger age than NF. Recent increases have higher proportion NF (as observed in adolescent samples). |
| de Graaf, N. M., Manjra, II, Hames, A., & Zitz, C. (2019). Thinking about ethnicity and gender diversity in children and young people. Clinical Child Psychology & Psychiatry, 24(2), 291–303. | 2019 | UK<br>Gender Identity Development Service, Tavistock, London | 2<br>Referrals only–no GD clinical verification | Black and minority ethnic groups were underrepresented. |
| De Pedro, K. T., Gilreath, T. D., Jackson, C., & Esqueda, M. C. (2017). Substance Use Among Transgender Students in California Public Middle and High Schools. The Journal of school health, 87(5), 303–309. | 2017 | US<br>2013–2015 California Healthy Kids Survey (CHKS) | 2<br>Survey. Self-identified as transgender. | Transgender youth at increased risk for substance misuse. |
| De Vries, A. L. C., Noens, I. L. J., Cohen-Kettenis, P. T., Van Berckelaer-Onnes, I. A., & Doreleijers, T. A. (2010). Autism spectrum disorders in gender dysphoric children and adolescents. Journal of Autism and Developmental Disorders, 40 (8), 930–936. | 2010 | NETHERLANDS<br>VU Medical Center, Amsterdam | 2<br>Not all participants diagnosed | Indication of association between GID and ASD |
| De Vries, A. L. C., McGuire, J. K., Steensma, T. D., Wagenaar, E. C. F., Doreleijers, T. A. H., & Cohen-Kettenis, P. T. (2014). Young adult psychological outcome after puberty suppression and gender reassignment. Pediatrics, 134(4), 696–704. | 2014 | NETHERLANDS<br>Center of Expertise on Gender Dysphoria (CEGD), Amsterdam | 0 | Included in Paper 1 (epidemiology) |
| Delahunt, J. W., Denison, H. J., Sim, D. A., Bullock, J. J., & Krebs, J. D. (2018). Increasing rates of people identifying as transgender presenting to Endocrine Services in the Wellington region. New Zealand Medical Journal, 131(1468), 33–42. | 2018 | NEW ZEALAND<br>Wellington Endocrine Service for Capital & Coast District Health Board | 1<br>No separate data on adolescents (includes ≥18 yrs) | Observed increase in referrals for people under 30, also increase in requests for female to male transition. |
| Drummond, K. D., Bradley, S. J., Peterson-Badali, M., VanderLaan, D. P., & Zucker, K. J. (2018). Behavior Problems and Psychiatric Diagnoses in Girls with Gender Identity Disorder: A Follow-Up Study. Journal of Sex & Marital Therapy, 44(2), 172–187. | 2018 | CANADA<br>Gender Identity Service, Center for Addiction and Mental Health, Toronto | 1, 6<br>Included some DSD; No separate data on adolescents | 12% (n = 3) of those referred in childhood showed persistent GID in adolescence / adulthood. Some indication of psychiatric vulnerability at follow-up, but a lot of variability. |
| Drummond, K. D., Bradley, S. J., Peterson-Badali, M., & Zucker, K. J. (2008). A follow-up study of girls with gender identity disorder. Developmental Psychology, 44(1), 34–45. | 2008 | CANADA<br>Gender Identity Service, Center for Addiction and Mental Health, Toronto | 1<br>Follow-up took place from age 17; small n under 18 (and no distinct data reported) | As above |

(*Continued*)

**Table 5.** (*Continued*)

| Reference | Date | Location & setting | Reason for exclusion | Notes |
|---|---|---|---|---|
| Durwood, L., McLaughlin, K. A., & Olson, K. R. (2017). Mental health and self-worth in socially transitioned transgender youth. Journal of the American Academy of Child & Adolescent Psychiatry, 56(2), 116–123. doi:10.1016/j.jaac.2016.10.016 | 2017 | NORTH AMERICA Multisite survey–TransYouth Project | 2 Survey data. Participants self-identify as transgender. | Socially transitioned young people had MH no worse than others their age |
| Getahun, D., Nash, R., Flanders, W. D., Baird, T. C., Becerra-Culqui, T. A., Cromwell, L.,. . . Goodman, M. (2018). Cross-sex Hormones and Acute Cardiovascular Events in Transgender Persons: A Cohort Study. Annals of Internal Medicine, 169(4), 205–213. doi:10.7326/M17-2785 | 2018 | USA Kaiser Permanente sites in Georgia, northern California, southern California | 1 Adults only | Indication of increased risk of cardiovascular events in transfeminine participants. |
| Handler, T., Hojilla, J. C., Varghese, R., Wellenstein, W., Satre, D. D., & Zaritsky, E. (2019). Trends in Referrals to a Pediatric Transgender Clinic. Pediatrics, 144(5), 11. | 2019 | USA Kaiser Permanente Northern California health system | 2 No clinical verification of GD / GID. | Observed increase in referrals in recent years. Large proportion identify as transmasculine. Treatment needs varied by age group. |
| Hannema, S. E., Schagen, S. E. E., Cohen-Kettenis, P. T., & Delemarre-Van De Waal, H. A. (2017). Efficacy and safety of pubertal induction using 17beta-estradiol in transgirls. Journal of Clinical Endocrinology and Metabolism, 102(7), 2356–2363. | 2017 | NETHERLANDS Centre of Expertise on Gender Dysphoria (CEGD), Amsterdam | 1 No separate data on adolescents (includes ≥18 yrs) | Estradiol effective for pubertal induction in NM. |
| Hughes, S. K., VanderLaan, D. P., Blanchard, R., Wood, H., Wasserman, L., & Zucker, K. J. (2017). The Prevalence of Only-Child Status Among Children and Adolescents Referred to a Gender Identity Service Versus a Clinical Comparison Group. Journal of Sex & Marital Therapy, 43(6), 586–593. | 2017 | CANADA Gender Identity Service, CYFS, Toronto | 2, 4 Mean age of all groups <12 yrs. No apparent clinical verification of GD / GID. | Prevalence of only-child status not elevated in gender-referred children (compared to other clinical populations). |
| Janssen, A., Huang, H., & Duncan, C. (2016). Gender Variance among Youth with Autism Spectrum Disorders: A Retrospective Chart Review. Transgender Health, 1(1), 63–68. | 2016 | USA New York University Child Study Center | 2 Derived from clinical ASD sample. No clinical verification of GD / GID. | Participants with ASD diagnosis more likely to report gender variance on CBCL then CBCL normative sample. |
| Jarin, J., Pine-Twaddell, E., Trotman, G., Stevens, J., Conard, L. A., Tefera, E., & Gomez-Lobo, V. (2017). Cross-sex hormones and metabolic parameters in adolescents with gender dysphoria. Pediatrics, 139 (5) (no pagination) (e20163173). | 2017 | USA Multi-site. MedStar Washington Hospital Center and Children's National Medical Center (both, Washington, DC). University of Maryland Medical Center, Baltimore. Cincinnati Children's Hospital Medical Center, Ohio. | 1 No separate data on adolescents (includes ≥18 yrs) | Testosterone use associated with increased hemoglobin and hematocrit, increased BMI, and lowered high-density lipoprotein levels. No significant change in those taking estrogen. |
| Jensen, R. K., Jensen, J. K., Simons, L. K., Chen, D., Rosoklija, I., & Finlayson, C. A. (2019). Effect of Concurrent Gonadotropin-Releasing Hormone Agonist Treatment on Dose and Side Effects of Gender-Affirming Hormone Therapy in Adolescent Transgender Patients. Transgender Health, 4(1), 300–303. | 2019 | USA Ann & Robert H. Lurie Children's Hospital of Chicago, Chicago, Illinois* | 0 | Included in Paper 1 (epidemiology) |

(*Continued*)

**Table 5.** (Continued)

| Reference | Date | Location & setting | Reason for exclusion | Notes |
|---|---|---|---|---|
| Joseph, T., Ting, J., & Butler, G. (2019). The effect of GnRH analogue treatment on bone mineral density in young adolescents with gender dysphoria: findings from a large national cohort. Journal of pediatric endocrinology & metabolism: JPEM., 31. | 2019 | UK<br>Gender Identity Development Service, Tavistock & UCLH Early Intervention programme @ national endocrine clinic, London | 0 | Included in Papers 1 (epidemiology) and 3 (treatment outcomes) |
| Kaltiala, R., Bergman, H., Carmichael, P., de Graaf, N. M., Egebjerg Rischel, K., Frisen, L.,. . . Waehre, A. (2020). Time trends in referrals to child and adolescent gender identity services: a study in four Nordic countries and in the UK. Nordic Journal of Psychiatry, 74(1), 40–44. | 2020 | DENMARK, FINLAND, NORWAY, SWEDEN, & the UK | 2<br>No clinical verification of GD / GID. | Comprehensive overview of referrals in 5 countries. Same pattern of increase, especially in NF, as in included papers. |
| Kaltiala, R., Heino, E., Tyolajarvi, M., & Suomalainen, L. (2020). Adolescent development and psychosocial functioning after starting cross-sex hormones for gender dysphoria. Nordic Journal of Psychiatry, 74(3), 213–219. | 2020 | FINLAND<br>Tampere University Hospital, Department of Adolescent Psychiatry | 1<br>Mean age at assessment 18.1 | MH problems persisted during treatment–concluded that GD treatment not enough to address MH problems. |
| Kaltiala-Heino, R., Tyolajarvi, M., & Lindberg, N. (2019). Gender dysphoria in adolescent population: A 5-year replication study. Clinical Child Psychology and Psychiatry, 24(2), 379–387. | 2019 | FINLAND<br>School survey in Tampere | 2<br>Survey data.<br>No clinical verification of GD / GID. | Apparent increase in likely clinically-significant GD in adolescent population (2013–2017). |
| Katz-Wise, S. L., Ehrensaft, D., Vetters, R., Forcier, M., & Austin, S. B. (2018). Family Functioning and Mental Health of Transgender and Gender-Nonconforming Youth in the Trans Teen and Family Narratives Project. Journal of sex research, 55(4–5), 582–590. | 2018 | USA<br>New England region–survey from range of services / organisations | 2<br>Survey data.<br>No clinical verification of GD / GID. | MH concerns reported. Better family functioning (from young person's persepective) associated with better MH outcomes. |
| Khatchadourian, K., Amed, S., & Metzger, D. L. (2014). Clinical management of youth with gender dysphoria in Vancouver. Journal of Pediatrics, 164(4), 906–911. doi:10.1016/j.jpeds.2013.10.068 | 2014 | CANADA<br>British Columbia<br>Children's Hospital Transgender Program, Vancouver | 1<br>No separate data on adolescents (includes ≥18 yrs) | Median age at initiation of testosterone NF 17.3 years (range 13.7–19.8 years); median age at initiation of estrogen in NM 17.9 years (range 13.3–22.3 years). Intervnention appropriate in selected individuals with relevant clinical support. |
| Klaver, et al. (2018). Early Hormonal Treatment Affects Body Composition and Body Shape in Young Transgender Adolescents. Journal of Sexual Medicine, 15(2), 251–260. | 2018 | NETHERLANDS<br>VU University Medical Center, Amsterdam (forerunner to CEGD) | 0 | Included in Papers 1 (epidemiology) and 3 (treatment outcomes) |
| Klaver, et al. (2020). Hormonal Treatment and Cardiovascular Risk Profile in Transgender Adolescents. Pediatrics, 145 (3), 03. | 2020 | NETHERLANDS<br>VU University Medical Center, Amsterdam (forerunner to CEGD) | 0 | Included in Papers 1 (epidemiology) and 3 (treatment outcomes) |
| Klein, D. A., Roberts, T. A., Adirim, T. A., Landis, C. A., Susi, A., Schvey, N. A., & Hisle-Gorman, E. (2019). Transgender Children and Adolescents Receiving Care in the US Military Health Care System. JAMA Pediatrics. | 2019 | USA<br>Military Health System Data Repository | 1<br>No separate data on adolescents (includes ≥18 yrs) | Increase in service use 2010–2017. Prescriptions increased with higher parental rank. |
| Kolbuck, V. D., Muldoon, A. L., Rychlik, K., Hidalgo, M. A., & Chen, D. (2019). Psychological functioning, parenting stress, and parental support among clinic-referred prepubertal gender-expansive children. Clinical Practice in Pediatric Psychology, 7 (3), 254–266. doi:10.1037/cpp0000293 | 2019 | USA<br>Division of Adolescent Medicine, Ann & Robert H. Lurie Children's Hospital of Chicago* | 4<br>Sample <12 yrs. | Association between GD symptoms in ADHD (hyperactive-impulsive) and CD where parenting stress high. |

*(Continued)*

**Table 5.** (Continued)

| Reference | Date | Location & setting | Reason for exclusion | Notes |
|---|---|---|---|---|
| Lawlis, S. M., Donkin, H. R., Bates, J. R., Britto, M. T., & Conard, L. A. E. (2017). Health Concerns of Transgender and Gender Nonconforming Youth and Their Parents Upon Presentation to a Transgender Clinic. Journal of Adolescent Health, 61(5), 642–648. doi:10.1016/j.jadohealth.2017.05.025 | 2017 | USA 'a transgender clinic at a large tertiary pediatric hospital in the Midwest.' Oklahoma University Children's Hospital* | 2 | 66.1% attending for first appointment NF |
| Lee, J. Y., Finlayson, C., Olson-Kennedy, J., Garofalo, R., Chan, Y.-M., Glidden, D. V., & Rosenthal, S. M. (2020). Low Bone Mineral Density in Early Pubertal Transgender/Gender Diverse Youth: Findings From the Trans Youth Care Study. Journal of the Endocrine Society, 4 (9), 1–12. doi:10.1210/jendso/bvaa065 | 2020 | USA Trans Youth Care Study: Children's Hospital Los Angeles, Lurie Children's Hospital, Boston Children's Hospital, and University of California San Francisco Benioff Children's Hospital | 0 | Included in Paper 3 (treatment outcomes) |
| Lobato, M. I., Koff, W. J., Schestatsky, S. S., Chaves, C. P. V., Petry, A., Crestana, T.,. . . Henriques, A. A. (2007). Clinical characteristics, psychiatric comorbidities and sociodemographic profile of transsexual patients from an outpatient clinic in Brazil. International Journal of Transgenderism, 10(2), 69–77. doi:10.1080/15532730802175148 | 2007 | BRAZIL Hospital de Clínicas de Porto Alegre | 1 | 42.7% had at least one psychiatric comorbidity |
| Lopez, et al. (2018). Trends in the use of puberty blockers among transgender children in the United States. Journal of Pediatric Endocrinology and Metabolism, 31(6), 665–670. | 2018 | USA US Pediatric Health and Information System (PHIS) database | 0 | Included in Paper 1 (epidemiology) and 3 (treatment outcomes) |
| Lothstein, L. M. (1980). The adolescent gender dysphoric patient: An approach to treatment and management. Journal of Pediatric Psychology, 5(1), 93–109. | 1980 | USA Case Western Reserve University (CWRU) Gender Identity Clinic, Cleveland, Ohio | 1, 3 | Case series from over 40 years ago |
| Lynch, M. M., Khandheria, M. M., & Meyer, W. J., III. (2015). Retrospective study of the management of childhood and adolescent gender identity disorder using medroxyprogesterone acetate. International Journal of Transgenderism, 16(4), 201–208. doi:10.1080/15532739.2015.1080649 | 2015 | USA Gender Identity Clinic, University of Texas Medical Branch | 3 Case series | Medroxyprogesterone Acetate found to be effective and low-cost oral alternative to injectable or implant GnRH analogues. Response to treatment and compliance were favourable. |
| Manners, P. J. (2009). Gender identity disorder in adolescence: A review of the literature. Child and Adolescent Mental Health, 14(2), 62–68. | 2009 | UK (review) Salomons Clinical Psychology Training Program, Canterbury Christ Church University | 5 Review | Review now out of date |
| May, T., Pang, K., & Williams, K. J. (2017). Gender variance in children and adolescents with autism spectrum disorder from the National Database for Autism Research. International Journal of Transgenderism, 18(1), 7–15. doi:10.1080/15532739.2016.1241976 | 2017 | USA National Database for Autism Research | 2 Derived from clinical ASD sample. No clinical verification of GD / GID. | Higher prevalence of gender variance in ASD sample compared to non-referred samples (but similar to other clinical samples). |
| Millington, K., Liu, E., & Chan, Y. M. (2019). The utility of potassium monitoring in gender-diverse adolescents taking spironolactone. Journal of the Endocrine Society, 3(5), 1031–1038. | 2019 | USA Gender Management Service Program, Boston Children's Hospital | 1 Sample likely to include those ≥18 yrs | Hyperkalemia in patients taking spironolactone for gender transition rare. Routine electrolyte monitoring may be unnecessary. |

(*Continued*)

**Table 5.** (*Continued*)

| Reference | Date | Location & setting | Reason for exclusion | Notes |
|---|---|---|---|---|
| Millington, K., Schulmeister, C., Finlayson, C., Grabert, R., Olson-Kennedy, J., Garofalo, R.,. . . Chan, Y. M. (2020). Physiological and Metabolic Characteristics of a Cohort of Transgender and Gender-Diverse Youth in the United States. Journal of Adolescent Health, 67(3), 376–383. | 2020 | USA Children's Hospital Los Angeles/University of Southern California, Boston Children's Hospital/Harvard Medical School, the Ann & Robert H. Lurie Children's Hospital of Chicago/Northwestern University, and the Benioff Children's Hospital/ University of California San Francisco | 4 Sample 1 aged 8–14 (so mostly under 12s); sample 2 aged 12–20 (so included over 18s). | Description of baseline metabolic characteristics–will be useful to see cohorts followed up. |
| Munck, E. T. (2000). A retrospective study of adolescents visiting a Danish clinic for sexual disorders. International Journal of Adolescent Medicine and Health, 12(2–3), 215–222. doi:10.1515/IJAMH.2000.12.2–3.215 | 2000 | DENMARK Sexological Clinic, Copenhagen University Hospital | 1 Mean age over 20 years | Description of cohort 1686–1995. Up to age 16, majority were NM. From age 17, majority were NF. |
| Nahata, L., Tishelman, A. C., Caltabellotta, N. M., & Quinn, G. P. (2017). Low Fertility Preservation Utilization Among Transgender Youth. Journal of Adolescent Health, 61(1), 40–44. | 2017 | USA Division of Endocrinology, Department of Pediatrics, Nationwide Children's Hospital, The Ohio State University College of Medicine, Columbus, Ohio* | 5 Same sample already described (Nahata et al, 2017) | See epidemiological data in main review (Nahata et al, 2017). |
| Neyman, A., Fuqua, J. S., & Eugster, E. A. (2019). Bicalutamide as an Androgen Blocker With Secondary Effect of Promoting Feminization in Male-to-Female Transgender Adolescents. Journal of Adolescent Health, 64(4), 544–546. | 2019 | USA Pediatric Endocrine Clinic, Riley Hospital for Children, Indiana | 1, 3 Where adolescent data separated, includes very small sample (case series) | Evidence that bicalutamide may be viable alternative to gonadotrophin-releasing hormone analogues in NM ready to transition |
| O'Bryan, J., Scribani, M., Leon, K., Tallman, N., Wolf-Gould, C., Wolf-Gould, C., & Gadomski, A. (2020). Health-related quality of life among transgender and gender expansive youth at a rural gender wellness clinic. Quality of Life Research, 29 (6), 1597–1607. | 2020 | USA The Gender Wellness Center (GWC) of the Bassett Health-care Network, New York | 1 Upper age limit 25 yrs. No meaningful separate data on those under 18 yrs. | Poor MH reported (relative to general population). Long term follow-up needed. |
| Olson, J., Schrager, S. M., Belzer, M., Simons, L. K., & Clark, L. F. (2015). Baseline Physiologic and Psychosocial Characteristics of Transgender Youth Seeking Care for Gender Dysphoria. Journal of Adolescent Health, 57(4), 374–380. doi:10.1016/j.jadohealth.2015.04.027 | 2015 | USA Center for Transyouth Health and Development, Children's Hospital Los Angeles, California | 1 | Awareness of gender incongruity from young age (mean 8.3 yrs). Physiological characteristics within normal ranges. 35% experiencing depression; 51% contemplated suicide; 30% attempted suicide. |
| Olson-Kennedy, J., Okonta, V., Clark, L. F., & Belzer, M. (2018). Physiologic Response to Gender-Affirming Hormones Among Transgender Youth. Journal of Adolescent Health, 62(4), 397–401. | 2018 | USA Center for Transyouth Health and Development, Children's Hospital Los Angeles, California | 1 | Use of gender affirming hormones not associated with clinically significant changes in metabolic parameters. May not need to frequently monitor transgender adolescents. |
| Olson-Kennedy, J., Warus, J., Okonta, V., Belzer, M., & Clark, L. F. (2018). Chest Reconstruction and Chest Dysphoria in Transmasculine Minors and Young Adults: Comparisons of Nonsurgical and Postsurgical Cohorts. JAMA Pediatrics, 172(5), 431–436. doi:10.1001/ jamapediatrics.2017.5440 | 2018 | USA Center for Transyouth Health and Development, Children's Hospital Los Angeles, California | 1 | Mean age at chest surgery 17.5 (2.4) years. 49% younger than 18 years. All postsurgical participants (n = 68) felt surgery had been a good decision. Loss of nipple sensation most common side-effect. |

(*Continued*)

**Table 5.** (Continued)

| Reference | Date | Location & setting | Reason for exclusion | Notes |
|---|---|---|---|---|
| Olson-Kennedy, J., Chan, Y. M., Garofalo, R., Spack, N., Chen, D., Clark, L.,. . . Rosenthal, S. (2019). Impact of Early Medical Treatment for Transgender Youth: Protocol for the Longitudinal, Observational Trans Youth Care Study. JMIR Research Protocols, 8(7), e14434. | 2019 | USA<br>Children's Hospital Los Angeles/ University of Southern California, Boston Children's Hospital/Harvard University, Lurie Children's Hospital of Chicago/Northwestern University, and the<br>Benioff Children's Hospital/ University of California San Francisco | 0 | Included in Paper 1 (epidemiology) |
| Ospina, N. M. S., Maraka, S., Rodriguez-Gutierrez, R., Davidge-Pitts, C. J., Nippoldt, T. B., & Murad, M. H. (2016). Effect of sex steroids on the bone health of transgender individuals: A systematic review and meta-analysis. Endocrine Reviews. Conference: 98th Annual Meeting and Expo of the Endocrine Society, ENDO, 37(2 Supplement 1). | 2016 | International (researcher based in USA) | 5<br>Systematic review | Bone mineral density (lumbar spine) increased in NM 12–24 months after initiating feminising hormone therapy. No changes in NF with masculinising therapy. |
| Pakpoor, J., Wotton, C. J., Schmierer, K., Giovannoni, G., & Goldacre, M. J. (2016). Gender identity disorders and multiple sclerosis risk: A national record-linkage study. Multiple Sclerosis, 22(13), 1759–1762. | 2016 | UK<br>English national Hospital Episode Statistics (HES) and mortality data | 5<br>No useable data for our questions | |
| Pang, K. C., de Graaf, N. M., Chew, D., Hoq, M., Keith, D. R., Carmichael, P., & Steensma, T. D. (2020). Association of Media Coverage of Transgender and Gender Diverse Issues With Rates of Referral of Transgender Children and Adolescents to Specialist Gender Clinics in the UK and Australia. JAMA Network Open, 3(7), e2011161. | 2020 | UK & Australia | 2<br>Data from referral only: GD not clinically verified. | Evidence of association between media coverage and number of new referrals to services. |
| Perez-Brumer, A., Day, J. K., Russell, S. T., & Hatzenbuehler, M. L. (2017). Prevalence and Correlates of Suicidal Ideation Among Transgender Youth in California: Findings From a Representative, Population-Based Sample of High School Students. Journal of the American Academy of Child & Adolescent Psychiatry, 56(9), 739–746. doi:10.1016/j.jaac.2017.06.010 | 2017 | USA<br>California Healthy Kids Survey | 2 | Transgender youth had 2.99 higher odds of reporting past-year suicidal ideation than non-transgender youth. |
| Perl, et al. (2020). Blood Pressure Dynamics After Pubertal Suppression with Gonadotropin-Releasing Hormone Analogs Followed by Testosterone Treatment in Transgender Male Adolescents: A Pilot Study. Lgbt Health, 7 (6), 340–344. | 2020 | ISRAEL<br>Gender Dysphoria Clinic at Dana-Dwek Children's Hospital, Gender Clinic, Tel Aviv Sourasky Medical Center | 0 | Included in Paper 3 (treatment outcomes) |
| Peterson, C. M., Matthews, A., Copps-Smith, E., & Conard, L. A. (2017). Suicidality, Self-Harm, and Body Dissatisfaction in Transgender Adolescents and Emerging Adults with Gender Dysphoria. Suicide & Life-Threatening Behavior, 47(4), 475–482. | 2017 | USA<br>Cincinnati Children's Hospital Medical Center Transgender Clinic, Ohio | 1 | 30.3% transgender youth reported history of ≥1 suicide attempt; 41.8% history self-injury. Higher suicidality in NF than NM. |

(*Continued*)

**Table 5.** (Continued)

| Reference | Date | Location & setting | Reason for exclusion | Notes |
|---|---|---|---|---|
| Quinn, V. P., Nash, R., Hunkeler, E., Contreras, R., Cromwell, L., Becerra-Culqui, T. A.,. . . Goodman, M. (2017). Cohort profile: Study of Transition, Outcomes and Gender (STRONG) to assess health status of transgender people. BMJ Open, 7(12), e018121. | 2017 | USA Kaiser-Permanente records, California and Georgia | 1 No useable data by age group | Useable data (proportion NM to NF) described in Becerra-Culqui et al (2018) |
| Reisner, S. L., Biello, K. B., Hughto, J. M. W., Kuhns, L., Mayer, K. H., Garofalo, R., & Mimiaga, M. J. (2016). Psychiatric diagnoses and comorbidities in a diverse, multicity cohort of young transgender women: Baseline Findings from Project LifeSkills. JAMA Pediatrics, 170(5), 481–486. | 2016 | USA: Chicago and Boston–Project LifeSkills | 1, 2 No separate data on adolescents (includes ≥18 yrs). Current GD not an inclusion criterion. | 41.5% of sample of NM had 1 or more mental health or substance dependence diagnoses; 20% had 2 or more comorbid psychiatric diagnoses |
| Reisner, S. L., Vetters, R., Leclerc, M., Zaslow, S., Wolfrum, S., Shumer, D., & Mimiaga, M. J. (2015). Mental health of transgender youth in care at an adolescent Urban community health center: A matched retrospective cohort study. Journal of Adolescent Health, 56(3), 274–279. | 2015 | US Sidney Borum Jr Health Center, Boston | 1 No separate data on adolescents (includes ≥18 yrs; mean age 19.6±3.0). | Increased risk of MH problems in transgender vs cisgender youth. No difference by natal sex. |
| Rider, G. N., Berg, D., Pardo, S. T., Olson-Kennedy, J., Sharp, C., Tran, K. M.,. . . Keo-Meier, C. L. (2019). Using the Child Behavior Checklist (CBCL) with transgender/gender nonconforming children and adolescents. Clinical Practice in Pediatric Psychology, 7(3), 291–301. doi:10.1037/cpp0000296 | 2019 | USA Trans Youth and Family Allies project (national) | 2 Survey | No significant impact in use of gendered scoring templates on CBCL |
| Roberts, A. L., Rosario, M., Slopen, N., Calzo, J. P., & Austin, S. B. (2013). Childhood gender nonconformity, bullying victimization, and depressive symptoms across adolescence and early adulthood: An 11-year longitudinal study. Journal of the American Academy of Child & Adolescent Psychiatry, 52(2), 143–152. doi:10.1016/j.jaac.2012.11.006 | 2013 | USA Growing Up Today Study (GUTS) (national) | 1, 2 Survey Age range 12–30 | Large longitudinal cohort. Association between gender nonconformity and depressive symptoms. |
| Röder, M., Barkmann, C., Richter-Appelt, H., Schulte-Markwort, M., Ravens-Sieberer, U., & Becker, I. (2018). Health-related quality of life in transgender adolescents: Associations with body image and emotional and behavioral problems. International Journal of Transgenderism, 19(1), 78–91. doi:10.1080/15532739.2018.1425649 | 2018 | GERMANY Hamburg Gender Identity Service for Children and Adolescents | 1 Upper age 18.2 | Health related quality of life (HRQoL) generally poorer in ttransgender adolescents vs normative scores. Body satisfaction and internalising problems significant predictors of HRQoL. |
| Schagen, S. E., Delemarre-van de Waal, H. A., Blanchard, R., & Cohen-Kettenis, P. T. (2012). Sibling sex ratio and birth order in early-onset gender dysphoric adolescents. Archives of Sexual Behavior, 41(3), 541–549 | 2012 | NETHERLANDS VU University Medical Center, Amsterdam (forerunner to CEGD) | 0 | Included in Paper 1 (epidemiology) |

(*Continued*)

**Table 5.** (Continued)

| Reference | Date | Location & setting | Reason for exclusion | Notes |
|---|---|---|---|---|
| Schagen, S. E. E., Cohen-Kettenis, P. T., Delemarre-van de Waal, H. A., & Hannema, S. E. (2016). Efficacy and Safety of Gonadotropin-Releasing Hormone Agonist Treatment to Suppress Puberty in Gender Dysphoric Adolescents. Journal of Sexual Medicine, 13(7), 1125–1132. | 2016 | NETHERLANDS Centre of Expertise on Gender Dysphoria (CEGD), Amsterdam | 1 Upper age 18.6 | Triptorelin effective in suppressing puberty. Routine monitoring of gonadotropins, sex steroids, creatinine, and liver function may not be necessary. |
| Schagen, et al. (2018). Changes in Adrenal Androgens During Puberty Suppression and Gender-Affirming Hormone Treatment in Adolescents With Gender Dysphoria. Journal of Sexual Medicine, 15 (9), 1357–1363. | 2018 | NETHERLANDS Centre of Expertise on Gender Dysphoria (CEGD), Amsterdam | 1 Max age 18.6 | No harmful effects of treatment of GnRHa and gender affirming hormone treatment on adrenal androgen levels were found during approximately 4 years of follow-up. |
| Schagen, et al. (2020). Bone Development in Transgender Adolescents Treated With GnRH Analogues and Subsequent Gender-Affirming Hormones. Journal of Clinical Endocrinology & Metabolism, 105(12), 01. | 2020 | VU University Medical Center, Amsterdam (forerunner to CEGD)* | 0 | Included in Paper 3 (treatment outcomes) |
| Shields, J. P., Cohen, R., Glassman, J. R., Whitaker, K., Franks, H., & Bertolini, I. (2013). Estimating population size and demographic characteristics of lesbian, gay, bisexual, and transgender youth in middle school. Journal of Adolescent Health, 52 (2), 248–250. | 2013 | USA Youth Risk Behavior Survey (YRBS), San Francisco, California | 2 Survey | 1.3% of middle school students identified as transgender |
| Shumer, D. E., Reisner, S. L., Edwards-Leeper, L., & Tishelman, A. (2016). Evaluation of Asperger Syndrome in Youth Presenting to a Gender Dysphoria Clinic. LGBT Health, 3(5), 387–390. | 2016 | USA Boston Children's Hospital | 1, 3 Sample included adults (max age 20 yrs). Small sub-sample: only 6 aged 12–18 yrs ASQ>80 | 9/39 (23.1%) GD participants had indication of Asperger Syndrome. |
| Skagerberg, E., Di Ceglie, D., & Carmichael, P. (2015). Brief Report: Autistic Features in Children and Adolescents with Gender Dysphoria. Journal of Autism and Developmental Disorders, 45(8), 2628–2632. | 2015 | UK Gender Identity Development Service, Tavistock, London | 2 No GD dx reported | Positive association between SRS scores and ASD symptoms. |
| Skagerberg, E., Parkinson, R., & Carmichael, P. (2013). Self-harming thoughts and behaviors in a group of children and adolescents with gender dysphoria. International Journal of Transgenderism, 14(2), 86–92. doi:10.1080/15532739.2013.817321 | 2013 | UK Gender Identity Development Service, Tavistock, London | 2 No GD dx reported | 24% self-harmed, 14% had thoughts of self-harming, suicide attempts indicated in 10% prior to attending GIDS. Thoughts of self-harm more common in NM, actual self-harm more common in NF. |
| Smith, Y. L. S., Van Goozen, S. H. M., & Cohen-Kettenis, P. T. (2001). Adolescents with gender identity disorder who were accepted or rejected for sex reassignment surgery: A prospective follow-up study. Journal of the American Academy of Child and Adolescent Psychiatry, 40(4), 472–481. | 2001 | NETHERLANDS University Medical Centre, Utrecht (moved to VUmc / CEGD in 2002). | 1 No separate data on adolescents (includes ≥18 yrs) | Group no longer GD after sex reassignment surgery. No one expressed regrets. Non-treated group showed some improvement in MH, but also had 'more dysfunctional psychological profile'. |
| Smith, Y. L. S., Van Goozen, S. H. M., Kuiper, A. J., & Cohen-Kettenis, P. T. (2005). Sex reassignment: Outcomes and predictors of treatment for adolescent and adult transsexuals. Psychological Medicine, 35(1), 89–99. | 2005 | NETHERLANDS VU University Medical Centre, Amsterdam (VUmc) or University Medical Centre, Utrecht (UMCU) | 1 No separate data on adolescents (includes ≥18 yrs) | Group no longer GD after sex reassignment surgery. |

(*Continued*)

**Table 5.** (Continued)

| Reference | Date | Location & setting | Reason for exclusion | Notes |
|---|---|---|---|---|
| Spack, N. P., Edwards-Leeper, L., Feldman, H. A., Leibowitz, S., Mandel, F., Diamond, D. A., & Vance, S. R. (2012). Children and adolescents with gender identity disorder referred to a pediatric medical center. Pediatrics, 129(3), 418–425. | 2012 | USA GeMS clinic, Endocrine Division, Children's Hospital Boston | 1 No separate data on adolescents (includes ≥18 yrs) | 44.3% had significant psychiatric history. Noted four-fold increase in presentations of GID following establishment of specialist service. |
| Steensma, T. D., & Cohen-Kettenis, P. T. (2015). More than two developmental pathways in children with gender dysphoria? Journal of the American Academy of Child and Adolescent Psychiatry, 54(2), 147–148. | 2015 | NETHERLANDS Centre of Expertise on Gender Dysphoria (CEGD), Amsterdam | 1 Letter to the editor: cannot determine if cohort already described in included papers. | Posits distinction between 'persisters' and 'persisters after interruption'. |
| Steensma, T. D., McGuire, J. K., Kreukels, B. P. C., Beekman, A. J., & Cohen-Kettenis, P. T. (2013). Factors associated with desistence and persistence of childhood gender dysphoria: A quantitative follow-up study. Journal of the American Academy of Child and Adolescent Psychiatry, 52(6), 582–590. | 2013 | NETHERLANDS Centre of Expertise on Gender Dysphoria (CEGD), Amsterdam | 1 No separate data on adolescents (includes ≥18 yrs) | Persistence of GD associated with early intensity of symptoms and being NF. Noted differing presentation by natal sex. |
| Steensma, T. D., Zucker, K. J., Kreukels, B. P. C., VanderLaan, D. P., Wood, H., Fuentes, A., & Cohen-Kettenis, P. T. (2014). Behavioral and emotional problems on the Teacher's Report Form: A cross-national, cross-clinic comparative analysis of gender dysphoric children and adolescents. Journal of Abnormal Child Psychology, 42(4), 635–647. doi:10.1007/s10802-013-9804-2 | 2014 | NETHERLANDS Centre of Expertise on Gender Dysphoria (CEGD), Amsterdam AND CANADA Gender Identity Service, CYFS, Toronto | 1 No separate data on adolescents (includes ≥18 yrs) (Adolescent subgroup likely to include those up to age 19–20, based on means / SDs given) | Teacher-reported emotional and behavioral problems greater in adolescents than in children. Internalising and externalising problems greater in NM than NF. Canadian sample had greater emotional and behavioural problems than Dutch sample. |
| Stoffers, I. E., de Vries, M. C., & Hannema, S. E. (2019). Physical changes, laboratory parameters, and bone mineral density during testosterone treatment in adolescents with gender dysphoria. Journal of Sexual Medicine, 16(9), 1459–1468. | 2019 | NETHERLANDS Department of Pediatrics, Leiden University Medical Centre, Leiden* | 0 | Included in Paper 3 (treatment outcomes) |
| Strang, J. F., Powers, M. D., Knauss, M., Sibarium, E., Leibowitz, S. F., Kenworthy, L.,. . . Anthony, L. G. (2018). "They Thought It Was an Obsession": Trajectories and Perspectives of Autistic Transgender and Gender-Diverse Adolescents. Journal of Autism and Developmental Disorders, 48(12), 4039–4055. | 2018 | USA Center for Neuroscience and Behavioral Medicine, Children's National Health System, Washington, DC* | 1 No separate data on adolescents (includes ≥18 yrs) | Useful qualitative study on young people's perspectives. No relevant data for our research questions. |
| Sumia, M., Lindberg, N., Tyolajarvi, M., & Kaltiala-Heino, R. (2016). Early pubertal timing is common among adolescent girl-to-boy sex reassignment applicants. European Journal of Contraception and Reproductive Health Care, 21(6), 483–485. | 2016 | FINLAND Tampere University Hospital, Tampere Helsinki University Hospital, Helsinki | 2 | GD in adolescence associated with early pubertal timing in NF |
| Sumia, M., Lindberg, N., Tyolajarvi, M., & Kaltiala-Heino, R. (2017). Current and recalled childhood gender identity in community youth in comparison to referred adolescents seeking sex reassignment. Journal of Adolescence, 56, 34–39. | 2017 | FINLAND School survey in Tampere & clinically referred population: Tampere and Helsinki | 1 No separate data on adolescents (includes ≥18 yrs) (Adolescent subgroup likely to include those 18+, based on means / SDs given) | Interesting exploration of gender identity in GD and community samples. No data directly relevant to our research questions. |

(*Continued*)

**Table 5.** (Continued)

| Reference | Date | Location & setting | Reason for exclusion | Notes |
|---|---|---|---|---|
| Tack, et al. (2018). Proandrogenic and Antiandrogenic Progestins in Transgender Youth: Differential Effects on Body Composition and Bone Metabolism. Journal of Clinical Endocrinology and Metabolism, 103(6), 2147–2156. | 2018 | BELGIUM Division of Pediatric Endocrinology, Ghent University* | 0 | Included in Papers 1 (epidemiology) and 3 (treatment outcomes) |
| Tollit, M. A., Pace, C. C., Telfer, M., Hoq, M., Bryson, J., Fulkoski, N.,. . . Pang, K. C. (2019). What are the health outcomes of trans and gender diverse young people in Australia? Study protocol for the Trans20 longitudinal cohort study. BMJ Open, 9 (11), e032151. | 2019 | AUSTRALIA Royal Children's Hospital Gender Service (RCHGS), Melbourne | 5 Cohort description | Protocol paper only. |
| Twist, J., & de Graaf, N. M. (2019). Gender diversity and non-binary presentations in young people attending the United Kingdom's National Gender Identity Development Service. Clinical Child Psychology and Psychiatry, 24(2), 277–290. | 2019 | UK Gender Identity Development Service, Tavistock, London | 2 No clinical verification of GD / GID–new questionnaire completed at presentation. | Useful in relation to prevalence of different types of gender self-identification at clinics |
| van der Miesen, A. I. R., Hurley, H., Bal, A. M., & de Vries, A. L. C. (2018). Prevalence of the Wish to be of the Opposite Gender in Adolescents and Adults with Autism Spectrum Disorder. Archives of Sexual Behavior, 47(8), 2307–2317. | 2018 | NETHERLANDS Centre of Expertise on Gender Dysphoria (CEGD), Amsterdam | 2 No clinical verification of GD / GID–endorsement of single item on YSR only. | Significantly more adolescents (6.5%) with ASD endorsed item expressing wish to be the opposite gender compared to the general population (3–5%). NF endorsed more then NM. Adolescents with ASD who endorsed gender item had higher YSR scores (poorer MH). No association with any specific subdomain of ASD. |
| van der Miesen, A. I. R., Steensma, T. D., de Vries, A. L. C., Bos, H., & Popma, A. (2020). Psychological Functioning in Transgender Adolescents Before and After Gender-Affirmative Care Compared With Cisgender General Population Peers. Journal of Adolescent Health, 66(6), 699–704. | 2020 | NETHERLANDS Centre of Expertise on Gender Dysphoria (CEGD), Amsterdam | 2 (group 1) 1 (group 2) | Poor MH in referrals. MH in those receiving treatment was similar to general population sample. |
| Van Donge, N., Schvey, N. A., Roberts, T. A., & Klein, D. A. (2019). Transgender Dependent Adolescents in the U.S. Military Health Care System: Demographics, Treatments Sought, and Health Care Service Utilization. Military medicine, 184(5–6), e447-e454. | 2019 | USA Transgender and gender-diverse clinic for children of military personnel | 1 No separate data on adolescents (includes ≥18 yrs) | Mean age at first gender-related visit 14.5 years (SD 3.2). History of self-harm (42%), suicidal ideation (70%), suicide attempt (21%), and psychiatric hospitalisation (33%). |
| Vlot, M. C., Klink, D. T., den Heijer, M., Blankenstein, M. A., Rotteveel, J., & Heijboer, A. C. (2017). Effect of pubertal suppression and cross-sex hormone therapy on bone turnover markers and bone mineral apparent density (BMAD) in transgender adolescents. Bone, 95, 11–19. | 2017 | NETHERLANDS Centre of Expertise on Gender Dysphoria (CEGD), Amsterdam | 1 No separate data on adolescents (includes ≥18 yrs) | Suppressing puberty by GnRHa leads to a decrease of bone turnover markers (BTMs) in transgender adolescents, but added value of evaluating BTMs in transgender adolescents seems to be limited and requires further research. DXA-scans remain important in follow-up. |
| Wallien, M. S. C., & Cohen-Kettenis, P. T. (2008). Psychosexual outcome of gender-dysphoric children. Journal of the American Academy of Child and Adolescent Psychiatry, 47(12), 1413–1423. | 2008 | NETHERLANDS VU University Medical Center (forerunner to CEGD), Amsterdam | 4 Onset <12 years | Most children with GD were not GD after puberty. Those with persistent GD had more intense GD in childhood than those desisting. |

(*Continued*)

**Table 5.** (Continued)

| Reference | Date | Location & setting | Reason for exclusion | Notes |
|---|---|---|---|---|
| Wallien, M. S. C., Swaab, H., & Cohen-Kettenis, P. T. (2007). Psychiatric comorbidity among children with gender identity disorder. Journal of the American Academy of Child and Adolescent Psychiatry, 46(10), 1307–1314. | 2007 | NETHERLANDS VU University Medical Center (forerunner to CEGD), Amsterdam | 4 all < 12 years | 52% of GID children had one or more other diagnoses. Internalising problems more common (37%) than externalising (23%). 31% of GID group had anxiety disorder. |
| Watson, R. J., Veale, J. F., & Saewyc, E. M. (2017). Disordered eating behaviors among transgender youth: Probability profiles from risk and protective factors. International Journal of Eating Disorders, 50(5), 515–522. doi:10.1002/eat.22627 | 2017 | CANADA Canadian Trans Youth Health Survey (national) | 2 Survey | High rates of eating disorder behaviour among self-identified transgender youth. Risk for eating disordered behaviours linked to enacted stigma and violence exposure, and offset by social supports. |
| Wood, H., Sasaki, S., Bradley, S. J., Singh, D., Fantus, S., Owen-Anderson, A.,. . . Zucker, K. J. (2013). Patterns of referral to a gender identity service for children and adolescents (1976–2011): age, sex ratio, and sexual orientation. Journal of Sex & Marital Therapy, 39(1), 1–6. | 2013 | CANADA Gender Identity Service, CYFS, Toronto | 1, 4 No separate data on adolescents (includes <12 yrs and ≥18 yrs) | Sharp increase in adolescent referrals in 2004–2007 time period (compared to 1976–2003), continued into 2008–2011 time block. NF exceeded NM in most recent (2008–2011) cohort. |
| Yadegarfard, M., Ho, R., & Bahramabadian, F. (2013). Influences on loneliness, depression, sexual-risk behaviour and suicidal ideation among Thai transgender youth. Culture, Health & Sexuality, 15(6), 726–737. doi:10.1080/13691058.2013.784362 | 2013 | THAILAND Survey through range of organisations, via Rainbow Sky Association, Bangkok | 1, 2 | Education level (did not graduate high school) associated with less loneliness but more depression than those with some university credit. |
| Zou, Y., Szczesniak, R., Teeters, A., Conard, L. A. E., & Grossoehme, D. H. (2018). Documenting an epidemic of suffering: low health-related quality of life among transgender youth. Quality of Life Research, 27(8), 2107–2115. | 2018 | USA Transgender Clinic of the Division of Adolescent and Transition Medicine, Cincinnati Children's Hospital Medical Center, Ohio | 1 | Transgender / gender non-conforming youth reported low health related quality of life across all domains. Most were significantly lower than healthy peers or peers with chronic diseases. |
| Zucker, K. J., Owen, A., Bradley, S. J., & Ameeriar, L. (2002). Gender-dysphoric children and adolescents: A comparative analysis of demographic characteristics and behavioral problems. Clinical Child Psychology and Psychiatry, 7(3), 398–411. doi:10.1177/1359104502007003007 | 2002 | CANADA Gender Identity Service, CYFS, Toronto | 1, 4 No separate data on adolescents (includes <12 yrs and ≥18 yrs) | 84.7% of adolescents had CBCL sum score in clinical range (>90th centile). Scores strongly predicted by peer relations scale (i.e., poor peer relations predicted behavioural psychopathology). |
| Zucker, K. J., Bradley, S. J., Owen-Anderson, A., Kibblewhite, S. J., & Cantor, J. M. (2008). Is gender identity disorder in adolescents coming out of the closet? Journal of Sex and Marital Therapy, 34(4), 287–290. | 2008 | CANADA Gender Identity Service, CYFS, Toronto | 1 | Same sample as Wood (2013) above. No new data relevant to our research questions. |
| Zucker, K. J., Bradley, S. J., Owen-Anderson, A., Kibblewhite, S. J., Wood, H., Singh, D., & Choi, K. (2012). Demographics, behavior problems, and psychosexual characteristics of adolescents with gender identity disorder or transvestic fetishism. Journal of Sex & Marital Therapy, 38(2), 151–189. | 2012 | CANADA Gender Identity Service, CYFS, Toronto | 1 | Percentage of youth with CBCL and YSR total scores in clinical range was similar to non-GID referred comparison group, higher than non-referred comparison group. |

(*Continued*)

**Table 5.** (*Continued*)

| Reference | Date | Location & setting | Reason for exclusion | Notes |
|---|---|---|---|---|
| Zucker, K. J., Bradley, S. J., Owen-Anderson, A., Singh, D., Blanchard, R., & Bain, J. (2010). Puberty-blocking hormonal therapy for adolescents with gender identity disorder: A descriptive clinical study. Journal of Gay & Lesbian Mental Health, 15(1), 58–82. doi:10.1080/19359705.2011.530574 | 2010 | CANADA Gender Identity Service, CYFS, Toronto | 1 | More likely to recommend puberty blockers for NF than NM, and less likely to recommend for young people with a lower YSR score. |

Key: * = derived from author's affiliation and description in paper.

ADHD: Attention Deficit / Hyperactivity Disorder; ASD: Autism Spectrum Disorder; ASQ: Asperger Syndrome Quotient; CBCL: Child Behavior Check List; CD: Conduct Disorder; DXA: Dual-energy X-ray Absorptiometry; GD: Gender Dysphoria; GID: Gender Identity Disorder; GnRHa: Gonadotropin-releasing hormone agonist; MH: Mental Health; NM / NF: Natal Male / Natal Female; SRS: Social Responsiveness Scale; yrs: years.

Exclusion codes: 1: Included ≥18 year olds–no distinct adolescent data; 2: Non clinically-verified GD; 3: Case study / series; 4: Only included <12 year olds; 5: No original data; 6: Non-GD population (e.g., LGBTQ); 7: Conference proceedings.

clear lack of research on GD in low and middle income countries in the scientific literature [86], so the impact of different service contexts in countries like India and Thailand cannot be properly considered [87].

## Conclusion

Adolescents presenting at specialist centres for support / treatment in relation to GD are highly likely to have other MH problems (including neurodivergence). If we want to develop a full understanding of individuals' needs, the quality, quantity of the scientific literature needs to improve, as does its representation of different global populations. It will be important to assess and record MH, including GD, in whole populations and to understand the complex contexts of young people's lived experiences. Clinical assessment needs to take a comprehensive approach and include specialists not only in GD but also in other relevant specialisms, such as neurodevelopmental disorders and eating disorders. The long-term outcomes for young people presenting to GD services, regardless of treatment decisions, need to be systematically recorded in an inclusive and representative way, including the use of qualitative methods to ensure young people's voices are not lost.

## Supporting information

**S1 Checklist.**
(DOCX)

## Acknowledgments

Special thanks to Ingrid Vinsa, Research Nurse and Administrator at the Gillberg Neuropsychiatry Centre, for her invaluable assistance in obtaining full text papers and assistance to CG in supervision of this piece of work.

## Author Contributions

**Conceptualization:** Lucy Thompson, Christopher Gillberg.

**Data curation:** Lucy Thompson, Darko Sarovic, Philip Wilson, Angela Sämfjord, Christopher Gillberg.

**Formal analysis:** Lucy Thompson.

**Investigation:** Lucy Thompson, Darko Sarovic, Philip Wilson, Angela Sämfjord, Christopher Gillberg.

**Methodology:** Lucy Thompson, Darko Sarovic, Philip Wilson, Angela Sämfjord, Christopher Gillberg.

**Project administration:** Lucy Thompson.

**Supervision:** Philip Wilson, Christopher Gillberg.

**Validation:** Darko Sarovic, Philip Wilson, Angela Sämfjord, Christopher Gillberg.

**Visualization:** Lucy Thompson.

**Writing – original draft:** Lucy Thompson, Christopher Gillberg.

**Writing – review & editing:** Lucy Thompson, Darko Sarovic, Philip Wilson, Angela Sämfjord, Christopher Gillberg.

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
