## [Decision Letter · Decision Letter 0]

18 Jan 2022

PGPH-D-21-00929

A PRISMA systematic review of adolescent gender dysphoria literature: 2) mental health

Dear Dr. Thompson,

Thank you for submitting your manuscript to PLOS Global Public Health. After careful consideration, we feel that it has merit but does not fully meet PLOS Global Public Health’s publication criteria as it currently stands. Therefore, we invite you to submit a revised version of the manuscript that addresses the points raised during the review process.

We look forward to receiving your revised manuscript.

Kind regards,

Runsen Chen

Academic Editor

Journal Requirements:

Additional Editor Comments (if provided):

Reviewers' comments:

Reviewer's Responses to Questions

**Comments to the Author**

1. Does this manuscript meet PLOS Global Public Health’s publication criteria? Is the manuscript technically sound, and do the data support the conclusions? The manuscript must describe methodologically and ethically rigorous research with conclusions that are appropriately drawn based on the data presented.

Reviewer #1: Yes

Reviewer #2: Yes

Reviewer #3: No

2. Has the statistical analysis been performed appropriately and rigorously?

Reviewer #1: N/A

Reviewer #2: Yes

Reviewer #3: No

3. Have the authors made all data underlying the findings in their manuscript fully available (please refer to the Data Availability Statement at the start of the manuscript PDF file)?

Reviewer #1: Yes

Reviewer #2: Yes

Reviewer #3: Yes

4. Is the manuscript presented in an intelligible fashion and written in standard English?

Reviewer #1: Yes

Reviewer #2: Yes

Reviewer #3: Yes

5. Review Comments to the Author

Reviewer #1: This was a pleasure to read. The manner through which the information is presented in a gender-sensitive manner was much appreciated. More specifically, your observation that the majority of participants in previous work identified as gender binary and that other identities remain under-represented frames the paper exactly where it needs to be.

Reviewer #2: This is a well-written, detailed paper outlining the findings of a systematic review on GD and mental health. It requires minor revisions to explain the eligibility and exclusion criteria. Some broader reflections on the state of the literature at the global level would also strengthen the paper.

In the Introduction, it would be helpful to situate the developments around GD within global health. I.e. in which settings are these debates and policies being established, and where is progress lagging? Can you provide broader context around how the issue is positioned at the global level?

On p. 3 (lines 69-71), there is a reference to ‘inherent bias’ in studies. Can you explain this further?

On p. 3, line 77, you reference ‘lack of good quality evidence’ – what evidence specifically is lacking? Do you mean evidence about the interaction between GD and mental health?

On p. 3, line 82, the reference to ‘the second of three sets of questions’ is confusing especially when followed by the list of seven questions (not appearing in sets) where Q4 is in bold. The explanation in lines 93-95 make sense but the initial explanation before this is not clear.

In your eligibility criteria you mention the huge amount of non-peer-reviewed literature. Can you say more about this? If not peer-reviewed, who is conducting/commissioning this body of literature? Was the grey literature primarily guidance materials or original research? Did the grey literature include research conducted in low and middle-income settings? Understanding the state of the non-peer reviewed literature is important in understanding why this literature was excluded. Could you please provide more justification to explain the decision to exclude grey lit especially given how lack of evidence is painted as a significant problem in the Introduction of the paper? This may need to be mentioned in the limitations.

On p. 5, line 137, you state that epidemiological, clinical and survey data are original data. Did you exclude qualitative studies, and if so, why were these studies excluded? This needs to be stated clearly as it sounds like qualitative studies were excluded. This may need to be mentioned in the limitations section of the paper.

Was any software used in screening?

p. 15, line 474 states that the quality of the literature needs to improve to understand the needs of individuals, but based on what has been stated in the paper, it is not simply about quality but also the quantity. There is an opportunity here to also draw attention to the geographical settings in which most research is occurring.

Reviewer #3: 1. why the title has number 2? Is it necessary to have 2 mental health? If the other papers are not necessarily related to the current review, I think it is not necessary to mention 2.

2. Mental health is a very broad area. It will be more helpful to have a specific area to review.

3. There are already existing reviews on mental health and GD. The authors mentioned they found "none". What are the specific researching strategies to find none?

4. The introduction did not clearly present the importance of the study. For example, why specific focus the adolescents, compared to adults?

5. In the method section, why chose 12-17 as adolescents? Need to list the criteria.

6. It is necessary to list the study types, and "not review" is not enough.

7. The searching terms are not adequate.

We will suggest the authors read more about how to conduct a systematic review and the standardised procedures.

6. PLOS authors have the option to publish the peer review history of their article (what does this mean?). If published, this will include your full peer review and any attached files.

**Do you want your identity to be public for this peer review?** For information about this choice, including consent withdrawal, please see our Privacy Policy.

Reviewer #1: **Yes: **Jasmin Lilian Diab

Reviewer #2: No

Reviewer #3: No

---

## [Decision Letter · Decision Letter 1]

6 Apr 2022

A PRISMA systematic review of adolescent gender dysphoria literature: 2) mental health

PGPH-D-21-00929R1

Dear Dr Thompson,

We are pleased to inform you that your manuscript 'A PRISMA systematic review of adolescent gender dysphoria literature: 2) mental health' has been provisionally accepted for publication in PLOS Global Public Health.

Best regards,

Runsen Chen

Academic Editor

Reviewer Comments (if any, and for reference):

Reviewer's Responses to Questions

**Comments to the Author**

1. If the authors have adequately addressed your comments raised in a previous round of review and you feel that this manuscript is now acceptable for publication, you may indicate that here to bypass the “Comments to the Author” section, enter your conflict of interest statement in the “Confidential to Editor” section, and submit your "Accept" recommendation.

Reviewer #2: All comments have been addressed

2. Does this manuscript meet PLOS Global Public Health’s publication criteria? Is the manuscript technically sound, and do the data support the conclusions? The manuscript must describe methodologically and ethically rigorous research with conclusions that are appropriately drawn based on the data presented.

Reviewer #2: Yes

3. Has the statistical analysis been performed appropriately and rigorously?

Reviewer #2: N/A

4. Have the authors made all data underlying the findings in their manuscript fully available (please refer to the Data Availability Statement at the start of the manuscript PDF file)?

Reviewer #2: Yes

5. Is the manuscript presented in an intelligible fashion and written in standard English?

Reviewer #2: Yes

6. Review Comments to the Author

Reviewer #2: Thanks for addressing the previous round of comments. The methods section is now much clearer in particular.

7. PLOS authors have the option to publish the peer review history of their article (what does this mean?). If published, this will include your full peer review and any attached files.

**Do you want your identity to be public for this peer review?** For information about this choice, including consent withdrawal, please see our Privacy Policy.

Reviewer #2: No
